# Antibiotic sensitivity reveals that wall teichoic acids mediate DNA binding during competence in *Bacillus subtilis*

Nicolas Mirouze [1,3], Cécile Ferret[1], Charlène Cornilleau[1,2] & Rut Carballido-López [1]

Despite decades of investigation of genetic transformation in the model Gram-positive bacterium *Bacillus subtilis*, the factors responsible for exogenous DNA binding at the surface of competent cells remain to be identified. Here, we report that wall teichoic acids (WTAs), cell wall-anchored anionic glycopolymers associated to numerous critical functions in Gram-positive bacteria, are involved in this initial step of transformation. Using a combination of cell wall-targeting antibiotics and fluorescence microscopy, we show that competence-specific WTAs are produced and specifically localized in the competent cells to mediate DNA binding at the proximity of the transformation apparatus. Furthermore, we propose that TuaH, a putative glycosyl transferase induced during competence, modifies competence-induced WTAs in order to promote (directly or indirectly) DNA binding. On the basis of our results and previous knowledge in the field, we propose a model for DNA binding and transport during genetic transformation in *B. subtilis*.

[1] MICALIS, INRA, AgroParisTech, Université Paris-Saclay, 78350 Jouy-en-Josas, France. [2] Inovarion, 75013 Paris, France. [3] Present address: Institute for Integrative Biology of the Cell (I2BC), INSERM, CEA, CNRS, Université Paris-Sud, Orsay, 91190 Gif sur Yvette, France. Correspondence and requests for materials should be addressed to N.M. (email: nicolas.mirouze@i2bc.paris-saclay.fr)

During evolution, the ability of bacteria to adapt to evolving environments often results from the acquisition of new genes, and therefore new functions, through Horizontal Gene Transfer (HGT). HGT, defined as the transmission of genetic material between organisms that are not in a parent-offspring relationship, ensures genomic plasticity and acquisition of pathogenicity islands and antibiotic resistances in bacteria[1]. HGT can occur naturally via three main mechanisms: transduction, conjugation and genetic transformation. The latter involves binding and transport of high molecular-weight exogenous DNA across the cell envelope (cell wall and membrane(s)), and homologous recombination with the chromosome of the recipient cell. Genetic transformation is the only HGT mechanism entirely directed by the recipient cell as all the proteins required are encoded by the core chromosome. Expression of genes essential for genetic transformation requires that bacterial cells enter a differentiated state called competence, which has been extensively studied in a number of bacterial species[2].

The model Gram-positive bacterium Bacillus subtilis has been used as a reference organism for the study of competence during decades. Competence development relies on an elaborate signal transduction system that senses environmental stimuli and transfers this information to the competence-specific transcriptional machinery. At the end of this signaling pathway the early transcriptional regulator ComK[3] activates the expression of around a hundred late competence genes, which include most of the genes involved in genetic transformation[4]. Despite decades of investigation, very little is known about the first extracellular steps of genetic transformation in B. subtilis. In particular, it is still unknown how exogenous DNA binds to the surface of competent cells. In Streptococcus pneumoniae, it was recently proposed that the type IV-like competence pseudopilus is the primary receptor of exogenous DNA during transformation[5]. However, Briley and co-workers showed that the pseudopilus has no DNA-binding role in B. subtilis[6]. These authors also showed that ComGA, a competence-induced cytoplasmic protein that forms membrane-associated polar clusters at the place where transforming DNA import occurs[7], is the only known protein absolutely required for DNA binding[6]. Therefore, the identity of the extracellular factor responsible for the initial binding of exogenous DNA at the surface of B. subtilis competent cells remains therefore unknown.

In Gram-positive bacteria, the cell wall consists of a thick layer of peptidoglycan (PG), a three-dimensional mesh of glycan chains cross-linked by short peptide bridges and functionalized with anionic glycopolymers named teichoic acids (TAs)[8]. TAs include both wall teichoic acids (WTAs), which are covalently attached to PG via disaccharide linkage units, and lipoteichoic acids (LTAs), which are anchored in the cytoplasmic membrane[8]. In B. subtilis, the genes encoding the proteins involved in WTAs synthesis are annotated as tag genes[9]. This pathway leads to the production, modification, export and anchoring to PG of glycerol phosphate repeats[10]. Cryo-electron microscopy images suggest that WTAs extend well beyond the PG, representing the outermost layer of the cell envelope exposed to the environment[11]. WTAs play numerous essential functions regulating cell morphology, cell division, autolytic activity, ion homeostasis, phage adsorption, and protection of the cell from host defenses[10]. WTAs are commonly decorated by D-alanyl esters[12] or glycosyl moieties[13]. Such tailoring modifications significantly affect WTAs physical properties and functions[10].

Under conditions of phosphate limitation, synthesis of WTAs is arrested and phosphate-free glycopolymers named teichuronic acids (TUAs)[14] are synthesized instead. This results from activation of the transcription of the tua operon (controlling TUAs synthesis) and repression of the transcription of the tag operon[15]. WTAs are subsequently released from the cell wall, degraded, and

the phosphate liberated from their degradation is taken up by the cell for other cellular processes. Meanwhile, TUAs replace WTAs in the cell wall, maintaining its global negative charge[16].

The use of antibiotics can provide important insights into the mechanisms underlying cellular processes. The effect of a range of antibiotics targeting different cellular functions (DNA, RNA, protein and cell wall synthesis) on the formation of competent B. subtilis cells was reported in a study from the early 80 s[17]. Interestingly, we noticed that two antibiotics targeting cell wall synthesis were reported to have opposite effects in this study: tunicamycin blocked genetic transformation, while methicillin had no effect[17]. Methicillin, an antibiotic from the widely used ß-lactam family, was known to inhibit PG cross-linking[18]. Tunicamycin, a glucosamine-containing antibiotic, was known to inhibit enzymes transferring hexose-1-phosphates to membrane-embedded lipid phosphates in both eukaryotes and prokaryotes[19]. In bacteria, it was thought to inhibit the initial membrane-bound reaction of PG synthesis catalyzed by MraY[20]. Since tunicamycin and methicillin had opposite effect, the authors of this study concluded that genetic transformation was dependent on the synthesis of PG but not on the final process of its cross-linking. However, it was later shown that in Gram-positive bacteria tunicamycin targets the biosynthetic pathways of both PG and surface glycopolymers (WTAs and TUAs)[21]. At low concentrations (<5 µg/ml) tunicamycin inhibits TagO, the enzyme that catalyzes the first step of WTAs and TUAs synthesis[21]. At higher concentrations (>10 µg/ml) tunicamycin additionally blocks MraY activity[20]. This prompted us to hypothesize that synthesis of surface glycopolymers, and not of PG, might be essential for genetic transformation. In addition, it was then tempting to speculate that WTAs or TUAs might be the missing extracellular factor involved in the initial DNA binding at the surface of B. subtilis competent cells.

Here, we investigated the effect of antibiotics targeting either PG or anionic glycopolymers synthesis on genetic transformation in B. subtilis. We show that tunicamycin blocks genetic transformation through the inhibition of TagO activity. Using targocil, an antibiotic that specifically targets the WTAs biosynthetic pathway of Staphylococcus aureus[22], we confirm that these PG-anchored glycopolymers (and not TUAs) are essential for genetic transformation in B. subtilis. Finally, using fluorescently-labeled DNA we demonstrate that DNA binding at the surface of competent cells is greatly affected both in tunicamycin-treated cells and in cells lacking TuaH, a putative glycosyl transferase encoded by the tua operon and specifically induced during competence. We propose a model in which WTAs specifically produced and modified during competence promote DNA binding, directly or indirectly, during genetic transformation in B. subtilis.

## Results

**Tunicamycin blocks genetic transformation.** The effect of diverse antibiotics on genetic transformation was previously investigated using a two-step transformation protocol that consists in sequentially culturing B. subtilis in two synthetic media[23,24]. This method confers an elevated transformation efficiency (>10⁻⁴, one cell out of ten thousand is transformed) after 90 min of growth in the second medium (Supplementary Fig. 1). The authors showed that addition of tunicamycin (5 µg/ml) strongly inhibited genetic transformation while addition of methicillin (0,1 µg/ml) had no effect[17]. We confirmed these results using the same two-step protocol, as well as a traditional one-step transformation protocol (Fig. 1a, b and Table 1). While the two cell wall antibiotics blocked vegetative growth, only tunicamycin inhibited transformation. To exclude the possibility that tunicamycin prevented the appearance of transformants by inhibiting

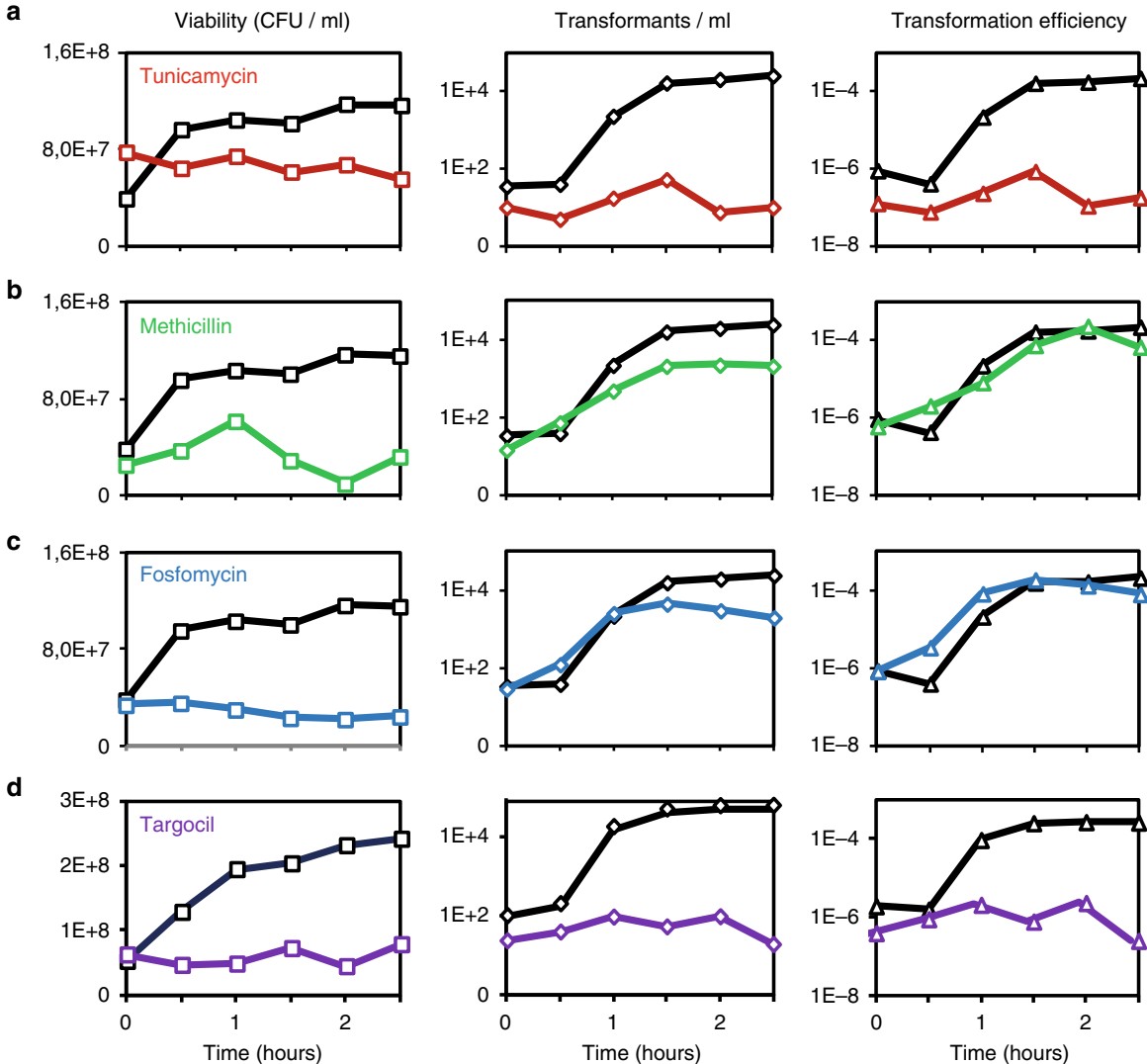

**Fig. 1** Effect of cell wall-targeting antibiotics on genetic transformation. For details of the experimental procedure, see Supplementary Fig. 1. When *B. subtilis* cells (wild-type strain 168 in panels **a**, **b**, **c** and strain NC288 ($\Delta tagGH^{Bs}$, $amyE$::P$_{hyperspank}$- $tarGH^{Sa}$) in panel **d**) reached stationary phase in SPI medium ($T_0$ in Fig. 1 and Supplementary Fig. 1), we diluted the culture in SPII medium. Expression of *tarGH* genes (NC288) was induced by addition of 1 mM IPTG, supplemented in SPI and SPII. Drugs were added (color lines), or not (black lines), at $T_0$. We used tunicamycin (a, red, 5 µg/ml), methicillin (**b**, green, 0.1 µg/ml), fosfomycin (**c**, blue, 320 µg/ml) and targocil (**d**, purple, 275 µg/ml). Viability and the number of transformants per ml were evaluated every 30 min by plating on the adequate selective media. Transformation efficiencies were calculated by dividing the number of transformants/ml by the viability at each time point

the development of competence, we used a transcriptional fusion between the promoter of *comGA* and the luciferase gene as a reporter for the expression of competence genes. We also used a strain natively expressing a *comK-gfp* fusion to quantify the percentage of competent cells. ComK, the master competence regulator[25], displays a homogeneous cytoplasmic localization in competent cells[26], which can then be readily distinguished from non-competent cells in the microscopy field. Addition of tunicamycin did not affect the expression of *comGA* or the percentage of competent cells in the culture (Supplementary Fig. 2a, b, respectively).

Tunicamycin can inhibit the synthesis of both PG and surface glycopolymers[21]. To determine whether inhibition of the PG biosynthetic pathway, of the WTAs/TUAs pathway, or of both, was responsible for blocking genetic transformation in tunicamycin-treated (5 µg/ml) cells, we first investigated the effect of fosfomycin. Fosfomycin is a broad-spectrum antibiotic that specifically targets MurA, the enzyme catalyzing the first

committed step in PG synthesis[27]. Like methicillin, addition of fosfomycin had no effect on genetic transformation (Fig. 1c). We concluded that PG synthesis is not essential for genetic transformation[28], and that tunicamycin probably targets surface glycopolymers biosynthesis to inhibit genetic transformation.

To confirm this hypothesis, we attempted to complement the low transformation efficiency of tunicamycin-treated cells by over-expressing *tagO* from the inducible hyperspank promoter. *tagO* mutants, which lack both WTA and TUA polymers, are viable but display severe growth and morphology defects[29], which are similar to the defects displayed by tunicamycin-treated cells in both LB and competence medium (Fig. 2a and Supplementary Fig. 3). Our *tagO* overexpression construct was able to complement the growth (Supplementary Fig. 3a, 3b) and morphology defects (Fig. 2a and Supplementary Fig. 3c) of *tagO* null mutant cells, indicating that it was functional. We then determined the lowest tunicamycin concentration that would provide a maximum inhibition of genetic transformation

**Table 1 Transformation efficiencies**

| *B. subtilis* strain | Tuni | Transformation method | Transformation efficiency | Standard deviation | Ratio | Fold difference |
|---|---|---|---|---|---|---|
| 168 | No | 1-step | 8,41E-6 | 8,63E-7 | 1,00 | 1 |
| 168 | Yes | 1-step | 3,47E-8 | 8,63E-9 | 0,004 | 242,4 |
| 168 | No | 2-step | 2,11E-4 | 6,04E-5 | 1,00 | 1 |
| 168 | Yes | 2-step | 1,47E-7 | 1,20E-7 | <0,001 | 1452,5 |
| NC227 (Δ*tagO*) | No | 2-step | <1E-9 | – | <0,001 | >1000 |
| NC228 (Phs-*tagO*) | No | 2-step | 1,41E-4 | 8,84E-5 | 0,669 | 1,5 |
| NC228 (Phs-*tagO*) −IPTG | Yes | 2-step | 1,99E-6 | 2,02E-6 | 0,009 | 105,7 |
| NC228 (Phs-*tagO*) +IPTG | Yes | 2-step | 6,28E-5 | 2,03E-5 | 0,298 | 3,4 |
| NC238 (Δ*tuaA*) | No | 2-step | 1,65E-4 | 1,00E-4 | 0,782 | 1,3 |
| NC239 (Δ*tuaF*) | No | 2-step | 3,65E-5 | 1,53E-5 | 0,173 | 5,8 |
| NC240 (Δ*tuaG*) | No | 2-step | 2,95E-5 | 1,33E-5 | 0,140 | 7,1 |
| NC241 (Δ*tuaH*) | No | 2-step | 2,59E-6 | 2,73E-6 | 0,012 | 81,2 |
| NC243 (ΔΔ*tuaH*, Phs-*tagO*) | No | 2-step | 2,99E-6 | 9,12E-7 | 0,014 | 70,6 |
| NC281 (Δ*tuaA, Phs-tuaH*) | No | 2-step | 1,57E-4 | 2,46E-5 | 0,745 | 1,3 |
| NC282 (Δ*tuaF, Phs-tuaH*) | No | 2-step | 1,46E-4 | 4,68E-5 | 0,693 | 1,4 |
| NC283 (Δ*tuaG, Phs-tuaH*) | No | 2-step | 1,74E-4 | 4,45E-5 | 0,828 | 1,2 |
| NC284 (Δ*tuaH, Phs-tuaH*) | No | 2-step | 1,71E-4 | 3,41E-5 | 0,812 | 1,2 |
| CCBS536(Δ*tagE*) | No | 2-step | 2,02E-4 | 5,21E-5 | 1 | 1 |
| CCBS487(Δ*dltA*) | No | 2-step | 1,64E-4 | 7,46E-5 | 0,777 | 1,3 |

The different *B. subtilis* strains studied were transformed (1 or 2-steps protocol, as indicated), in the presence (0.31 μg/ml) or in the absence of tunicamycin, using chromosomal DNA carrying a tetracycline marker. Each data point was repeated at least 3 times. The last columns show the ratio and fold-difference between each data point and the wild-type reference

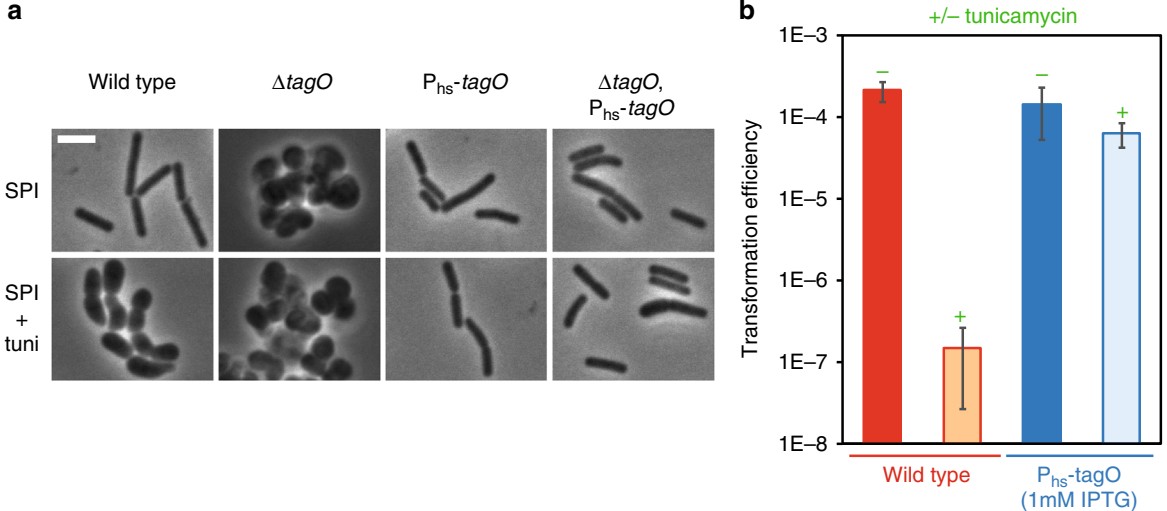

**Fig. 2** Tunicamycin targets TagO to inhibit genetic transformation. **a** The P_hs-*tagO* construct complements the morphological defects associated to either *tagO* deletion or addition of tunicamycin. Representative phase contrast images of cells of various genetic backgrounds (WT, strain 168; Δ*tagO*, strain NC227; P_hs-*tagO*, strain NC228; Δ*tagO* P_hs-*tagO*, strain NC229) exponentially growing in SPI (top panels) or in SPI supplemented with tunicamycin (0.31 μg/ml, bottom panels). In the *tagO* null mutant (strain NC227), the absence of WTAs induces important morphological defects leading to aggregated round cells[29]. When *tagO* is expressed from the hyperspank promoter (Phs), in the absence of *tagO* at the native locus (NC229), wild-type morphology is restored. In SPI supplemented with tunicamycin (0.31 μg/ml), wild-type cells (168) display growth defects comparable to those of the *tagO* mutant strain (NC227) in SPI (even though the cells are not as round and aggregated). Finally, expression of *tagO* from the hyperspank promoter restores normal cell morphology in the presence of tunicamycin. That result is true whether *tagO* is present (NC228) or not (NC229) at the native locus. Note that when tagO is inactivated at the native locus (NC229), very few cells look larger, potentially revealing an intermediate complementation. Complementary data are provided in Supplementary Fig. 3. Scale bar, 4 μm. **b** Transformation efficiency of the wild-type strain (168, red) and the *tagO* overexpression strain (NC228, blue) in the absence (−) or in the presence (+, 0.31 μg/ml) of tunicamycin. Strain NC228 was grown in the presence of 1 mM IPTG to induce *tagO* expression. Each transformation efficiencies presented correspond to the average of transformation efficiencies obtained in three independent experiments (biological replicates), also providing the standard deviation

(Supplementary Fig. 4). At this tunicamycin concentration (0.31 µg/ml), the associated morphological and growth phenotypes were also complemented by overexpression of *tagO* (Fig. 2a and Supplementary Fig. 3b). Finally, we verified the impact of *tagO* overexpression on genetic transformation using the optimized tunicamycin concentration. As expected, *tagO* overexpression restored an almost wild-type transformation efficiency in tunicamycin-treated cells (Fig. 2b).

Taken together, our results show that tunicamycin, a WTA/TUA-targeting antibiotic, inhibits genetic transformation. Remarkably, we were not able to detect any natural transformation using a *tagO* null mutant strain (Table 1). However, the very important growth defect associated to *tagO* deletion (Supplementary Fig. 3a) forced us to be cautious with this result.

**WTAs synthesis is essential for genetic transformation.** We next performed a time series experiment where we added the tunicamycin at different time points (i.e., every 30 min following dilution in SPII). We reasoned that if WTA/TUAs biosynthetic pathway was active in competent cells, then the timing of drug addition would influence the outcome of our experiments. As expected, the latter we added the antibiotic, the more transformants were detected (Supplementary Fig. 5), confirming that transformation depends on active synthesis of surface glycopolymers (WTAs and/or TUAs) in competent cells.

TagO participates in the synthesis of both WTAs and TUAs in *B. subtilis*[29]. To determine whether WTAs or TUAs were involved in genetic transformation, we investigated the effect of another antibiotic, named targocil, which specifically inhibits WTAs synthesis in *Staphylococcus aureus*[22]. Targocil was designed as a potent inhibitor of TarG, a transmembrane component of the ABC transporter that exports WTAs from the cytoplasm to the cell surface in *S. aureus* (TarGH$^{Sa}$)[22]. The *B. subtilis* homologs of this two component ABC transporter, TagGH$^{Bs}$, are not susceptible to targocil[30]. However, a *B. subtilis* strain expressing *tarGH$^{Sa}$* in a Δ*tagGH$^{Bs}$* background becomes

susceptible to this *S. aureus*-specific antibiotic[30]. Interestingly, in this complemented genetic background, addition of targocil significantly decreased the transformation efficiency (Fig. 1d). Taken together, these results confirm that WTAs synthesis is active in competent cells and that WTAs are essential for genetic transformation in *B. subtilis*.

**DNA binding is affected in tunicamycin-treated cells.** Because of their extracellular localization, we hypothesized that WTAs could be involved in exogenous DNA binding at the surface of competent cells. DNA binding and transport during transformation were recently investigated in *Helicobacter pylori* using fluorescently-labeled DNA[31]. We used a similar approach to evaluate, at the single cell level, binding of DNA at the surface of *B. subtilis* cells natively expressing the *comK-gfp* fusion as competence marker (Supplementary Fig. 6a). DNA binding was found to be highly specific to competent cells (Fig. 3 and Supplementary Fig. 6a). While 58.5% of competent cells (*n* > 400) displayed at least one focus of fluorescently-labeled DNA bound to their surface, only 2.4% of non-competence cells showed bound DNA (*n* > 2000). Among the competent cells associated to DNA, 64% displayed one focus of fluorescently-labeled DNA at their surface, 33% displayed two and 3% displayed three or more. DNA molecules where preferentially found at the vicinity of the cell poles (Supplementary Fig. 6a, c), as reported in a previous study[7]. We found 64% of bound DNA molecules within 0.5 µm from a cell pole (average cell length ~3.5 µm). Finally, deletion of *comGA* (the only gene previously known to be essential for DNA binding[6]) prevented DNA binding at the surface of competent cells (Supplementary Fig. 6b) demonstrating that DNA binding is dependent on competence development. When tunicamycin was added to the culture, the percentage of competent cells associated to DNA drastically decreased from 58.5 to 3.3% (*n* > 400) (Fig. 3a, c, Supplementary Fig. 7), indicating that DNA binding was inhibited. These results supported our hypothesis that WTAs are involved, directly or indirectly, in DNA binding at the surface of

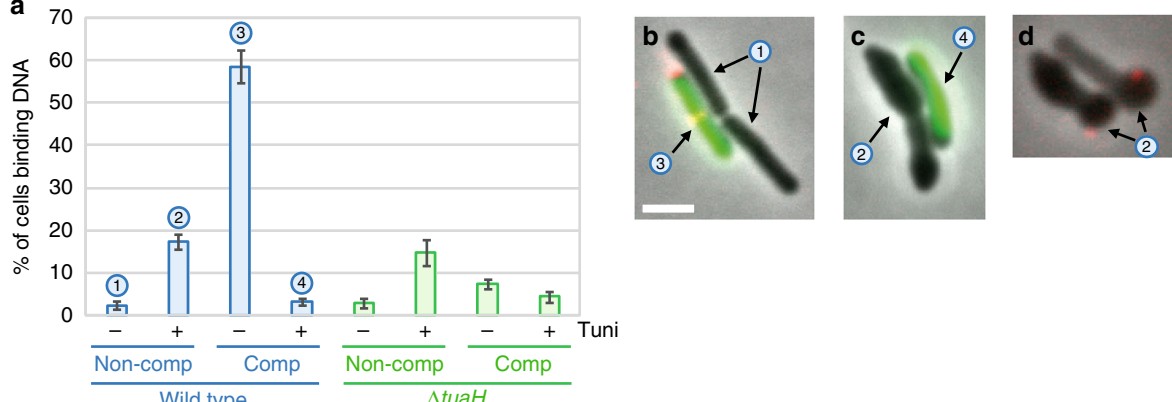

**Fig. 3** Visualization of exogenous DNA binding. Strains NC59 (wild-type, P$_{comK}$-comK-gfp) and NC275 (Δ*tuaH*, P$_{comK}$- comK-gfp) were grown using the two-step transformation protocol described in Supplementary Fig. 1. After dilution and 120 min of growth in SPII medium in the absence or in the presence of 5 µg/ml tunicamycin, fluorescently-labeled DNA (ATTO550-dUTP) was added to the cultures for 8 min. Cells were then washed in fresh SPII medium and observed under the microscope. Complementary data are provided in Supplementary Fig. 6. **a** Percentage of competent and non-competent cells of the wild-type (NC59, blue bars) and the *tuaH* mutant (NC275, green bars) genetic backgrounds binding DNA in the absence (−) or in the presence (+) of 5 µg/ml tunicamycin. Each experiment was repeated three times with more than 2000 (competent) and 400 (non-competent) cells analyzed per experiment. The percentage shown corresponds to the average of the percentages obtained in the three independent experiments. The standard deviation is also provided in each case. The numbers above the bars refer to the example cells shown in panels. **b–d** Overlay of representative phase contrast and their corresponding fluorescence images (ComK-GFP in green, DNA in red) of untreated (**b**) and tunicamycin-treated (**c**, **d**) wild-type cells (NC59). b, in the absence of tunicamycin both non competent (1) and competent (3) cells display normal morphologies but only competent cells bind DNA. **c**, **d**, in the presence of tunicamycin non competent cells (2) display abnormal morphologies and DNA binding is promoted at their surface (**d**), while DNA binding is lost at the surface of competent cells (4). Scale bar, 2 µm

*B. subtilis* competent cells. Noteworthy, the percentage of non competent cells binding DNA increased significantly in the presence of tunicamycin relative to untreated cells (from 2.4% to 17.5%) (Fig. 3a), suggesting that unlike competence WTAs, vegetative WTAs prevent DNA binding to the cell surface (see below).

**Local enrichment of WTAs at the transformation apparatus.** We next compared the distribution of WTAs and bound DNA at the surface of single competent cells. In a first attempt we used fluorescently-labeled DNA, as above, and fluorescein-labeled Concanavalin A (fl-ConA), which is a lectin commonly used to probe the location of WTAs in the surface of Gram-positive bacteria[32]. ConA specifically labels glucose residues decorating WTA ($\alpha$-GlcNAcylated WTAs,[33]). In agreement with this, a fl-ConA signal was detected in the surface of wild-type cells but not of cells lacking *tagE*, the gene responsible for WTA glycosylation in *B. subtilis*[33] (Supplementary Fig. 8). Unfortunately, WTAs/ DNA colocalization experiments were more complex than expected. When we allowed fluorescently-labeled DNA to bind to the cells first and then performed the fl-ConA labeling, the washes required to remove the excess of fl-ConA washed away an important fraction of bound DNA. At the opposite, when we first labeled WTAs with fl-ConA and then exposed the cells to fluorescently-labeled DNA, almost no binding was detected, suggesting that DNA could not access WTAs decorated with ConA. Like most lectins, ConA is a homotetramer[34] in which each sub-unit is composed of 235 amino acids and an $\alpha$-Glc binding site. Such structure could act as a barrier and prevent the access of DNA.

We therefore used an alternative, indirect way to compare DNA binding and WTAs distribution at the cell surface using ComGA clusters as a spatial reference. ComGA exhibits two distinguished localizations in *B. subtilis* competent cells[7]. First, when competence develops, ComGA is diffuse in the cytoplasm. Later, ComGA accumulates in membrane-associated polar clusters together with other competence proteins to form the transformation apparatus[7]. In some cells, and in addition of the polar clusters, ComGA can also display a distributed punctuate localization pattern, composed of 1 to 10 additional foci[7]. Using fluorescence microscopy colocalization, ComGA clusters were previously shown to correlate with DNA binding at the surface of competent cells in *B. subtilis*[7].

We quantified the distribution of fluorescently-labeled DNA relative to ComGA clusters using a strain natively expressing a *comGA-gfp* fusion (Supplementary Fig. 6c). When we plotted the distance between bound DNA and the closest ComGA cluster against the length of the cell ($n = 250$), a clear bias of DNA binding in the proximity of ComGA clusters was observed (Fig. 4a). The average distance between ComGA foci and bound DNA was around $0.68 \pm 0.30\,\mu m$ regardless the length of the cell. This finding confirmed that exogenous DNA is preferentially found in the vicinity of ComGA polar clusters.

We next compared the distribution of fl-ConA staining to the localization of ComGA clusters using a strain expressing a *comGA-rfp* fusion from the native *comGA* promoter. Strikingly, fl-ConA staining was much more intense at the surface of competent cells relative to non-competent cells (Fig. 4b and Supplementary Fig. 8a), suggesting that competent cells display more WTAs that bind ConA on their surface. This result is in agreement with our finding that active synthesis of WTAs occurs in competent cells (Figs. 1a, 2b and Supplementary Fig. 5), even though we cannot exclude that the accessibility of ConA to WTAs is increased in competent cells. Because ConA binds to $\alpha$-GlcNAcylated WTAs, it is also plausible that this increase in fl-

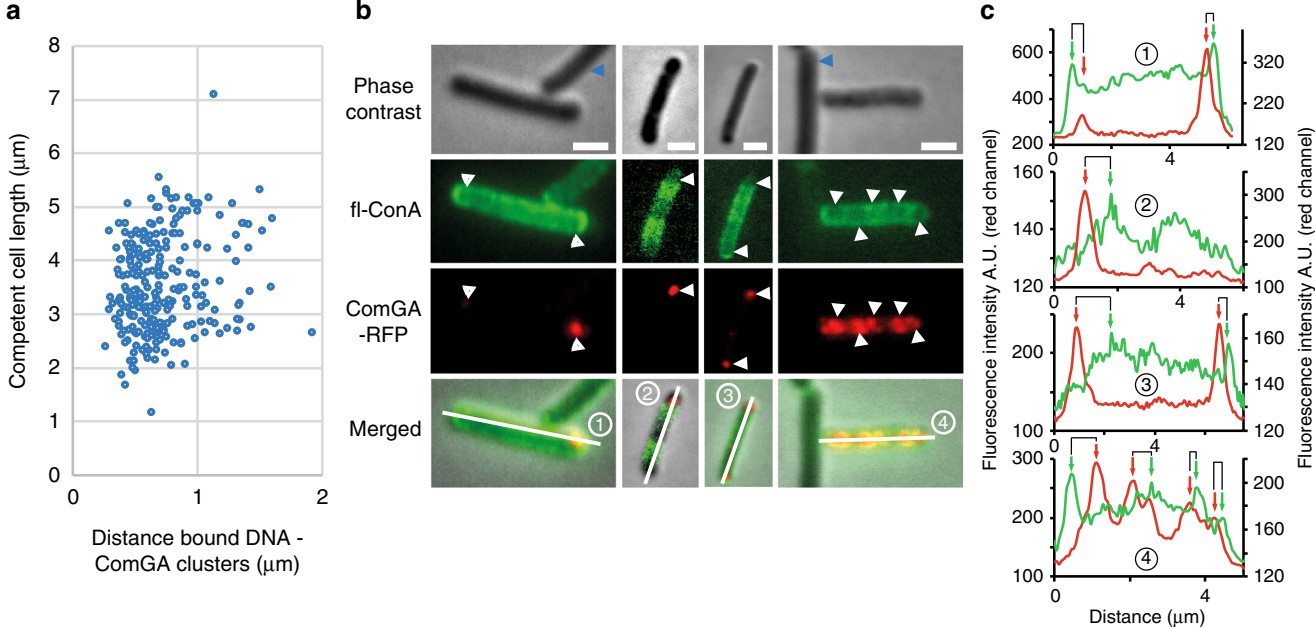

**Fig. 4** WTAs distribution at the surface of competent cells. Strains NC58 (*amyE*::P*comGA*-*comGA-gfp*) and NC118 (*thrC*::P*comGA*-*comGA-mrfpruby*) were used to compare the distribution of ComGA and DNA binding (**a**) and of ComGA and WTAs (**b** and **c**), respectively, in competent cells. **a** Distance between bound DNA (red channel) and ComGA foci (green channel) plotted against the average cell length (strain NC58, *n* > 250). **b** fl-ConA and ComGA dual labeling (strain NC118). For each field of view, phase contrast, GFP, RFP, and merged images are shown. White arrowheads point to cell locations where ComGA forms bright clusters. The fluorescence intensity profiles (green and red channels) along the midline of the four cells numbered in the merged images are shown in panel **c**. Blue arrowheads point to non-competent cells. Scale bar, 1 µm. **c** Green (fl-ConA) and red (ComGA-RFP) fluorescence intensity (A. U.) distributions along the midline of the four cells shown in panel **b** (numbered 1 to 4). Green arrows, local increases in fl-ConA intensity (peaks in the green curve). Red arrows, closest ComGA cluster to each local increase in fl-ConA staining (peaks in the red curve)

ConA labeling results from increased glycosylation of WTAs in competent cells. Interestingly, a ComGA cluster was present near every local increase in WTAs concentration (red and green peaks, respectively, in the intensity profiles in Fig. 4b, c). The average distance between WTAs maxima and the closest ComGA foci was 0,71 ± 0,33 μm ($n > 100$). This distance was comparable to the mean distance measured between bound DNA foci and ComGA clusters. We concluded that competent cells locally enrich their walls with WTAs, in particular around the transformation apparatus. This WTAs localization pattern would promote DNA binding in the close vicinity of the transformation apparatus and favor its transfer to the uptake machinery.

**Vegetative WTAs inhibit DNA binding.** Our finding that addition of tunicamycin increased the percentage of non-competent cells that bind DNA was intriguing because tunicamycin had the opposite effect in competent cells (Fig. 3a). While only 2.4% of untreated non-competent cells displayed bound DNA at their surface, this percentage reached 17.5% when cells were grown in the presence of tunicamycin (Fig. 3a). Unlike competent cells, non-competent cells keep growing and elongating after dilution into the second medium[35]. As a consequence, inhibition of WTAs synthesis by tunicamycin in actively growing cells leads to morphological defects. Altogether, these results suggested that WTAs decorating the surface of vegetatively growing cells prevent DNA binding and that a second DNA binding site, independent from competence and hidden in the cell wall, is revealed upon inhibition of WTAs synthesis. In support of this hypothesis, DNA bound at the surface of tunicamycin-treated vegetative (non-competent) cells preferentially localized to bulging areas (Fig. 3d), which lacked WTAs as shown by fl-ConA staining (Supplementary Fig. 9).

We next investigated the DNA binding properties of *tagO* mutant cells under growth conditions that do not support competence development. We grew Δ*tagO* mutant cells to exponential phase in rich LB medium and observed that these cells frequently bound exogenous DNA (Supplementary Fig. 10a), while only 2% of wild-type cells growing in the same conditions bound DNA (Supplementary Fig. 10b). It is important to mention that it is not possible to evaluate the number of DNA foci and the outline of individual cells (and thus the percentage of cells binding DNA) because of the severe morphological defects and the three-dimensional structure of the aggregates formed by *tagO* mutant cells[29] (Supplementary Fig. 10a).

Taken together, these results suggest that vegetative WTAs prevent DNA binding at the surface of vegetatively growing *B. subtilis*. A second type of DNA binding site may be buried in the cell wall, not exposed to the cell surface when vegetative WTAs are present but made accessible at sites devoid of WTAs such as cell bulges in tunicamycin-treated cells. Furthermore, to explain the different DNA-binding properties of vegetative WTAs (Fig. 3 and Supplementary Fig. 10) and WTAs produced during competence (Fig. 3 and Supplementary Fig. 6), we postulated that their respective structure or physiochemical properties must be different.

**Competence WTAs must be modified to allow DNA binding.** Could competence-induced WTAs display a tailoring modification that would determine their DNA-binding properties? Interestingly, *tuaF* and *tuaG*, two genes of the TUAs operon in *B. subtilis*, were reported to be induced during competence development in a genome-wide transcriptomic study[4]. Such specific induction of the expression of these two genes, located in sixth and seventh position, respectively, in the eight-gene containing *tua* operon (*tuaABCDEFGH*), suggested the presence of an

internal promoter right upstream of *tuaF*, in addition to the promoter present in front of the operon (Supplementary Fig. 11). This internal promoter would drive the expression of the three last genes of the operon, *tuaF*, *tuaG,* and *tuaH*, when competence develops. Premature termination of mRNA processing of the corresponding transcript could explain why *tuaH* was not reported to be upregulated like *tuaF* and *tuaG*, bringing its upregulation ratio below the threshold used in the study[4]. Consistent with the presence of such internal promoter, specific induction of expression of *tuaF, G,* and *H* has also been observed during sporulation[36]. To confirm the overexpression of *tuaH* during competence we used the luciferase from *Photinus pyralis* as a transcriptional reporter, as in ref. [37]. *tuaF* and *tuaH* were found to be overexpressed in competent cells while no expression could be detected for *tuaA* (Supplementary Fig. 12). These results showed that expression of the three last genes of the *tua* operon is specifically induced during competence, probably from a promoter present in front of *tuaF*.

Very little is known about the TUA biosynthesis pathway but it is currently proposed that *tuaG* and *tuaH* encode sugar transferases[16] while *tuaF* encodes a protein of unknown function. Specific expression of the end of the *tua* operon during competence raised the attractive possibility that one or several of these genes could catalyze the competence-specific modification of WTAs that determines their DNA binding properties. In Gram-positive bacteria, glycosylation is an important tailoring modification of WTAs that has been shown to play an important role in phage absorption[13], resistance to β-lactam antibiotics[38] and activation of the human complement system[39]. To investigate this hypothesis, we evaluated the transformation efficiency of strains in which *tuaA*, the first gene of the operon, and then the *tuaF, tuaG, or tuaH* genes were disrupted by plasmid insertion. Insertion of plasmid pIK156 in *tuaA, F, and G* had a limited impact on genetic transformation (1.3, 5.8, and 7.1-fold decrease, respectively) (Table 1). However, disruption of *tuaH* led to a 80-fold decrease of the transformation efficiency (Table 1). We postulated that the phenotypes associated to *tuaA*, *F* and *G* inactivation were potentially due to a polar effect on *tuaH* expression. In addition, because the *tua* operon is located immediately upstream of the *tagO* gene (Supplementary Fig. 11), a polar effect of *tuaH* inactivation on *tagO* expression could not be excluded. To test this, we overexpressed *tuaH* in the four mutant strains (*tuaA*, *F*, *G*, and *H*). Ectopic overexpression of *tuaH* restored a wild-type transformation efficiency in the *tuaH* mutant strain, indicating that the effect of *tuaH* inactivation on genetic transformation was direct, excluding a polar effect on *tagO*. Consistent with this finding, overexpression of *tagO* in the *tuaH* mutant did not complement its transformation efficiency (Table 1). *tuaH* overexpression also complemented the transformation defects of the *tuaA*, *F* and *G* mutant strains (Table 1), confirming that *tuaH* is the only gene of the *tua* operon required for genetic transformation in *B. subtilis*.

Next, we wondered if we could correlate the decrease of transformation efficiency associated to *tuaH* inactivation to a potential reduction in DNA binding in competent cells. Strikingly, only 7.5% of *tuaH* competent cells were associated with fluorescently-labeled DNA (Fig. 3a) in comparison to nearly 60% of wild-type competent cells. In the presence of tunicamycin, the percentage of competent and non-competent cells binding DNA was unaffected in cells lacking *tuaH* relative to wild-type cells (Fig. 3a). We concluded that *tuaH* plays a key role in DNA binding at the surface of *B. subtilis* competent cells.

Finally, we wondered if genes known to be involved in the modification of vegetative WTA could also be important, alongside TuaH, to promote DNA binding at the surface of competent cells. D-alanylation and glycoslisation are the major

modifications known to affect the properties and functions of WTAs[10]. We then investigated the effect of the deletion of *tagE* (WTA glycosylation,[13]) and of *dltA* (WTA alanylation,[12]) during competence. Deletion of *tagE* or *dltA* did not affect DNA binding (Supplementary Fig. 13) or transformation efficiency (Table 1). Thus, TuaH is currently the only candidate that could modify WTA in order to promote exogenous DNA binding during genetic transformation in *B. subtilis*.

## Discussion

Here, we show that WTAs, presumably modified by the putative sugar transferase TuaH, enable exogenous DNA binding, directly or indirectly, at the surface of *B. subtilis* competent cells. We have therefore identified both an essential missing factor involved in the initial binding of DNA during genetic transformation and a new function for WTAs in bacterial cell physiology.

WTAs have been shown to play a role in a large number of vital cellular functions in Gram-positive bacteria[10]. WTAs are anionic glycopolymers that display a large range of tailoring modifications thought to regulate their physical properties and functions[10]. In *Bacillus*, WTAs known modifications include D-alanylation[12] and glycosidic substitutions[13,40,41]. The main chain of WTAs is negatively charged because of the presence of phosphate, but the overall glycopolymer charge is generally altered (usually zwitterionic) by the groups attached to the WTAs backbone[10]. However, the extreme structural diversity of WTAs and their modifications remains largely unexplored due to their complex nature and current limits in glycosylation chemistry. We have found that WTAs specifically produced during competence are essential for exogenous DNA binding, while WTAs on the surface of vegetative cells have the opposite effect and prevent DNA binding. A simple explanation would be that in vegetative cells WTAs are anionic, and therefore promote surface exclusion of the also negatively charged DNA molecules. In contrast, our data suggest that in competent cells WTAs display tailoring modifications that change their structure or physicochemical properties, resulting in a polymer with different biological activity that promotes DNA binding at the cell surface. Because of the bistable nature of competence in *B. subtilis* (only 5% of cells in the population enter this developmental process), surface exclusion of DNA by non-competent cells might be an efficient way to concentrate DNA around the competent cells equipped for binding.

Expression of the *tuaH* gene, encoding a putative glycosyltransferase, is specifically induced during competence[4]. TuaH has been proposed (although experimental evidence was not provided) to link glucuronate to N-acetylgalactosamine in order to constitute the repeated unit forming TUAs in *B. subtilis*[16]. Here, we show that TuaH is required for DNA binding during competence independently of TUAs synthesis. Indeed, deletion of *tuaA*, encoding the first essential enzyme of the TUAs pathway, had no effect on genetic transformation. We postulate that TuaH modifies WTAs specifically produced during competence through a tailoring glycosylation that remains to be identified, to promote the binding of exogenous DNA. TuaH-added residues on WTAs of competent cells could directly bind DNA, affect the physicochemical or structural properties of the surrounding PG matrix, or recruit a third partner to promote DNA binding. Interestingly, glucose residues on WTAs have been shown to act as phage absorption primary receptors (see below)[42–44]. Glycosylation of WTAs would therefore appear as a crucial mechanism for HGT of virulence and resistance genes during both transduction and genetic transformation.

Importantly, ConA has been shown to specifically bind glycoslylated (α-GlcNAcylated) WTAs[32], and we found that fl-ConA staining was more intense at the surface of competent cells relative to non-competent cells. Furthermore, fl-ConA labeling showed a local enrichment at the vicinity of ComGA clusters. Thus, competent cells have more glycosylated WTAs than vegetative cells on their surface, and these become concentrated near the transformation apparatus. We then confirmed that exogenous DNA preferentially binds near ComGA clusters, as previously reported[7], and concluded that DNA binding is favored at competence-induced WTAs-rich regions near the transformation apparatus. Ultimately, this spatially optimized binding would increase the probability that the transforming DNA is transferred to the uptake machinery. In agreement with this model, Dubnau and coworkers proposed that ComGA interacts with and localizes the cytoplasmic moiety of an unknown DNA receptor[6] to explain that ComGA is essential for DNA binding and that transforming DNA preferentially binds next to ComGA clusters[7]. In this scenario, ComGA could interact with TuaH or other WTA synthetic proteins to specifically localize modified WTAs around the transformation apparatus. Interestingly, ComGA has also been shown to be essential for DNA transport, probably through the synthesis of the so-called pseudopilus[6]. The large number of activities associated to ComGA is of high interest in the field and places this cytoplasmic protein as an essential spatial and temporal regulator of the extracellular steps of genetic transformation.

However, because of experimental limitations (30 min between the addition of DNA to the cells and their observation under the microscope) we cannot demonstrate if DNA preferentially first binds next to the poles. An alternative hypothesis would be that DNA binding follows a mechanism similar to that used for bacteriophage SPP1 binding to *B. subtilis* cells. SPP1 scans the surface of *B. subtilis* cells using glycosylated WTAs as reversibly binding receptors, until it ultimately finds and irreversibly binds to its receptor YueB at the cell pole[45]. Similarly, transforming DNA might scan the surface of competent cells through reversible binding to glycosylated WTAs, which become enriched at the vicinity of the transformation apparatus. This unbiased binding would ensure a local increase of exogenous DNA concentration around competent cells and ultimately favor the otherwise low probability that DNA binds near the transformation apparatus. Once transferred to the uptake machinery, DNA binding would become irreversible and DNA would be transported across the cell envelope.

Antibiotics are powerful tools for uncovering the mechanisms underlying cellular processes. Part of our conclusions were reached using tunicamycin, an antibiotic that specifically inhibits WTAs synthesis at low concentrations[21]. Using our two-step transformation protocol, non-competent (vegetative) cells grown in the presence of tunicamycin displayed important morphological defects. At low tunicamycin concentration, PG synthesis continued while WTAs were diluted from the cell surface, leading to WTAs-free bulging regions. Strikingly, exogenous DNA specifically bound to these WTAs-free regions, suggesting that a secondary unidentified DNA binding site (buried in the cell wall of untreated vegetative cells) had been made accessible (Fig. 5). Consistent with these findings, a DNA binding site independent from competence development was previously observed in *Streptococci* vegetative cell walls[46]. In contrast, tunicamycin-treated competent cells did not display cell shape defects, a result consistent with the inhibition of sidewall elongation known to occur in cells that enter the competence state[26,28]. Our results that methicillin and fosfomycin had no effect on genetic transformation are also in agreement with PG synthesis being inactive during genetic competence. Such sidewall elongation inhibition in competent cells might prevent the dilution of vegetative WTAs when WTA synthesis is inhibited by tunicamycin, explaining why the second DNA binding site was not revealed in tunicamycin-treated competent cells (Fig. 5).

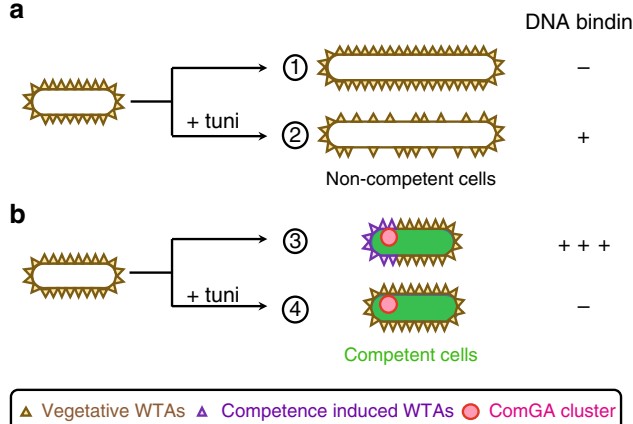

**Fig. 5** Model for binding of exogenous DNA at the surface of *B. subtilis* cells. Schematic illustration of the differential distribution of vegetative WTAs and competence-induced WTAs at the surface of non-competent (**a**) and competent (**b**) cells, in the absence or in the presence of tunicamycin. **a** Non competent cells elongate and divide in the absence of tunicamycin, synthesizing new vegetative WTAs that decorate the cell surface and prevent DNA binding (1). In the presence of tunicamycin, synthesis of vegetative WTAs is inhibited and cells elongate anarchically (2). This creates regions in the cell surface where WTAs are diluted or absent, revealing a low affinity DNA binding site. **b** Cells developing competence do not elongate. In the absence of tunicamycin they synthesize competence-specific tailored WTAs, which promote high affinity DNA binding at the vicinity of ComGA clusters (3). In the presence of tunicamycin, such competence-specific tailored WTAs are not synthesized preventing high affinity DNA binding. Because vegetative WTAs are not diluted, they keep preventing low affinity DNA binding at the cell surface (4)

In conclusion, this work provides strong evidence for competence-induced WTAs to mediate, directly or indirectly, the binding of exogenous DNA to the surface of *B. subtilis* competent cells. It is plausible that this mechanism is widespread in Gram-positive bacteria, as it has been demonstrated for the requirement of glycosylated WTAs as absorption phage receptor[42–44]. This role has been nevertheless attributed to the competence type IV-like pseudopilus in *S. pneumoniae*[5], suggesting that different strategies may have been developed to ensure DNA binding in different environmental conditions. Further studies will be required to investigate if WTAs are also involved in DNA binding in other competent Gram-positive bacteria, and to explore if lipopolysaccharides or other cell surface components could play a similar role in naturally competent Gram-negative bacteria. Elucidating the reaction catalyzed by TuaH, identifying the tailoring modification decorating competence-induced WTAs and understanding how it modulates WTAs structure and/or function are also exciting questions for future collaborations between biochemists, biophysicists, geneticists and structural biologists. Moreover, these studies should shed further light on the mechanisms of HGT and on the development of WTA-specific antibodies for vaccination purposes[47].

## Methods

**Microbiological methods**. *Bacillus subtilis* strains were constructed by natural genetic transformation with selection for the appropriate antibiotic resistance marker. For transformation, competent cultures were prepared and incubated in competence medium with transforming DNA (~1 μg/ml) for 30 min at 37 °C[48]. When needed, *B. subtilis* chromosomal DNA was prepared as detailed in ref. [49]. Transformants were selected using 10 μg/ml tetracycline, 100 μg/ml spectinomycin, 10 μg/ml kanamycin, 5 μg/ml chloramphenicol, 16 μg/ml phleomycin and 1 μg/ml erythromycin. The construction of the strains generated in this study is described below. All new constructs were sequenced after introduction in the *B. subtilis* chromosome.

*B. subtilis* strains were then grown in SPI, SPII[23,24], competence[48], or LB[50] media. High Mg$^{2+}$ concentration (20 mM) allows us to manipulate cell wall-related mutants (such as the Δ*tagO* mutant) the experiments (on plate or liquid over-night cultures).

*B. subtilis* strains used in this study are listed in Supplementary Table 1. The sequences of oligonucleotides used are listed in Supplementary Table 2.

**Construction of *amyE*::P$_{hs}$-*tagO***. To clone the *tagO* gene under the control of the IPTG-inducible hyperspank promoter (P$_{hs}$) at the ectopic *amyE* locus, the Gibson method[51] was used to join three fragments corresponding to the upstream (amy front) and downstream (amy back) regions of the *amyE* gene and to the *tagO* coding sequence. The amy front fragment also contained the hyperspank promoter and a kanamycin resistance cassette. These three fragments were amplified using primers amy-F and pDP111-MCS-R (amy front), amy-R and pDP111-MCS-F (amy back), and MCS-RBS-tagO-F and MCS-tagO-R (*tagO* gene), respectively. The amy front and back fragments were amplified from plasmid pDP111[52]. The *tagO* gene was amplified from chromosomal DNA of strain 168[53]. The three fragments were joined to produce the PCR product «amy-front-Phs-*tagO*-amy back» used to transform the wild-type strain (168), inserting the Phs-*tagO* construct, at the *amyE* locus and selecting for kanamycin resistance.

**Construction of *amyE*::P$_{hs}$-*tuaH***. The method was comparable to that described for the *amyE*::P$_{hs}$-*tagO* construct. The Gibson method[51] was used to join three fragments corresponding to the upstream (amy front) and downstream (amy back) regions of the *amyE* gene and the *tuaH* coding sequence. The amy front fragment also contained the hyperspank promoter and a kanamycin resistance cassette. The three fragments were amplified using primers amy-F and pDP111-MCS-R (amy front), amy-R and pDP111-MCS-F (amy back) and MCS-RBS-tuaH-F and MCS-tuaH-R (*tuaH* gene), respectively. The amy front and back fragments were amplified from plasmid pDP111[52]. The *tuaH* gene was amplified from chromosomal DNA of strain 168[53]. The three fragments were joined to produce the PCR product « amy-front-Phs-*tuaH*-amy back », used to transform the wild-type strain (168), inserting the Phs-*tuaH* construct, at the *amyE* locus and selecting for kanamycin resistance.

**Construction of pIK156-*tuaA*, -*tuaF*, -*tuaG*, and -*tuaH***. To generate gene disruption vectors, a PCR product containing a 530 to 1100 bp fragment of the *tuaA*, *tuaF*, *tuaG*, and *tuaH* genes was amplified from *B. subtilis* 168 chromosomal DNA using primer pairs sacI-tuaA and bamH1-tuaA, sacI-tuaF and bamH1-tuaF, sacI-tuaG and bamH1-tuaG, and sacI-tuaH and bamH1-tuaH, respectively. The PCR products were digested with SacI and BamHI and cloned into SacI and BamHI sites of plasmid pIK156[54], which carries a spectinomycin-resistance marker, generating plasmids pIK156-*tuaA*, pIK156-*tuaF*, pIK156-*tuaG*, and pIK156-*tuaH*. Insertion of the these plasmids in the targeted regions by single cross-over, led to the inactivation of the corresponding *tua* genes.

**P-tuaA, P-tuaF and P-tuaH-luc transcriptional fusions**. 1 kbp fragments ending right before the RBS of *tuaA*, *tuaF* and *tuaH* were amplified using primers 382-PtuaA_PtsI_fw / 383-PtuaA_KpnI_rev, 384-PtuaF_PtsI_fw / 385-PtuaF_KpnI_rev or 386-PtuaH_PtsI_fw / 387-PtuaH_KpnI_rev, respectively, and digested by KpnI and PstI. The pUC18cm-luc plasmid[55] was also digested with the same enzymes. Plasmid and fragments were ligated, and ligation mixtures were used to transform in electrocompetent Dh5α *Escherichia coli* cells. Single cross-over integration at the native loci was checked by PCR. All constructs were sequenced.

**One and two-step *B. subtilis* transformation protocols**. For strain construction purposes and where indicated in Table 1, we used a traditional one-step transformation protocol. Briefly, strains were grown to stationary phase for 5 h in competence medium (CM,[48]). Chromosomal DNA (1 μg/ml) was added to 1 ml of the culture for 30 min at 37 °C under agitation followed by 10 min in the presence of DNase to degrade any DNA that was not transformed during the first 30 min.

A two-step transformation protocol was used to test the effect of diverse antibiotics and gene deletions on genetic transformation. Our protocol was similar to what was described in[23,24]. *B. subtilis* strains were first grown in synthetic SPI medium. Upon entry in stationary phase, this first culture was diluted 10 times in fresh SPII medium supplemented with antibiotics where indicated ($T_0$). This second growth phase was then conducted for up to 2.5 h. Chromosomal DNA (1 μg/ml) harboring a tetracycline antibiotic resistance cassette was added to an aliquot of the culture for 30 min. After 10 min of DNase treatment, cells were washed once with unsupplemented SPII medium.

**Transformation efficiency measurements**. *B. subtilis* strains were transformed using chromosomal DNA of strain BD5004 carrying a tetracycline marker[56]. Cells were plated on LB plates for viability calculation and on LB plates supplemented with tetracycline to evaluate the number of transformants. Transformation efficiencies were calculated by dividing the number of transformants per ml by the viable count of each strain.

**Fluorescent-DNA (ATTO-550-DNA) preparation and labeling**. For microscopic visualization of exogenous DNA binding, we used a fluorescently-labeled 1,5 kb PCR fragment corresponding to the amplification of the *amyE* gene. PCR reactions were performed as explained in ref. [31]. Briefly, we used 1 µl of ExTaq enzyme from Takara in 100 µl on genomic DNA (20 ng) with 1 µM of each primer, 0.25 µM of each dNTP and 0.1 µM of ATTO550-aminoallyl-dUTP. Elongation was performed at 72 °C for 3 min per kb. Fragments were purified using Wizard SV Gel and PCR clean-up systems (Promega corporation).

200 ng of ATTO550-labeled DNA was mixed with 100 µl of cells grown for 2 h in SPII medium. The suspension was incubated for 8 min at 37 °C. Cells were then pelleted, washed once with 100 µl of fresh SPII medium and resuspended in 50 µl of fresh SPII medium.

**Concanavalin A (fl-ConA) labeling**. After 2 h of growth in SPII medium, 100 µl of cells were centrifuged and washed three times in PBS. Cells were resuspended in 100 µl of PBS supplemented with 1 mM $MnCl_2$, 1 mM $CaCl_2$ and 2 mg/ml of concanavalin labeled with fluorescein to produce the fluorescent derivative (fl-ConA, Sigma). Cells labeling was performed during 10 min at 37 °C under constant shaking. Cells were washed three times in PBS and observed under the microscope.

**Fluorescence microscopy**. Samples for microscopic observation were taken and immobilized on 1% agarose-coated microscope slides. Bacteria were imaged using an inverted microscope (Nikon Ti-E) equipped with a 100× oil immersion objective and an environmental chamber maintained at 37 °C. Conventional epi-fluorescence images were recorded using the 472/30-nm excitation filter and 520/35-nm emission filter for GFP, and the 562/40-nm excitation filter and 641/75-nm emission filter for RFP with an ORCA-R2 camera (Hamamatsu). Images were processed with NIS-Elements (Nikon) software. Exposure time was set up to 200 ms for GFP and 500 ms for RFP and fluorescein.

The average cell length measurements (Fig. 4a) were performed manually on the phase contrast images using the Fiji software. The fluorescence intensity profiles shown in Fig. 4c were generated using the plot profile function in the Fiji software.

**Luciferase assay**. Luciferase experiments were carried out as previously described[37,56]. In brief, the high instability of the luciferase allowed us to approach the measurement of a rate of expression[37], with a relatively small contribution from the cumulative effect of transcription. For detection of luciferase activity, strains were first grown in LB medium to an optical density at 600 nm ($OD_{600nm}$) of 2. Cells were then pelleted and resuspended in fresh SPI medium, adjusting all the cultures to an $OD_{600nm}$ of 2. These pre-cultures were then diluted 20-fold in fresh SPI medium and grown to stationary phase at 37 °C under shaking. SPI cultures were then diluted 10 fold in SPII medium (supplemented or not with tunicamycin, 5 µg/ml) and 200 µl of each culture were distributed in two wells in a 96-well black plate (PerkinElmer). 10 µl of luciferin (PerkinElmer) was added to each well to reach a final concentration of 1.5 mg/ml (4.7 mM). The cultures were incubated at 37 °C with agitation in a PerkinElmer Envision 2104 Multilabel Reader equipped with an enhanced sensitivity photomultiplier for luminometry. The temperature of the clear plastic lid was maintained at 38 °C to avoid condensation. Relative Luminescence Units (RLU) and $OD_{600nm}$ were measured at 2 min intervals. Data were plotted as RLU/OD (luminescence readings corrected for the OD) versus time from inoculation.

## Data availability

The data that support the findings of this study are available within the paper and its Supplementary Information files, or from the corresponding author upon request.

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

## Acknowledgements

We thank Stéphanie Marsin for providing ATTO-labeled PCR products and protocols used for preliminary experiments, and Suzanne Walker and Leigh Matano for kindly sharing targocil and *B. subtilis* strains. This work was supported by a "Return Post-doctoral grant" from the French National Research Agency to N.M. (ANR-12-PDOC-002, Cytostat) and a starting grant from the European Research Council to R.C.-L. (ERCStG 311231, BaCeMo).

## Author contributions

N.M. and R.C.-L. designed the study; N.M., C.F., and C.C. undertook experiments and performed analysis; N.M. and R.C.-L. drafted and wrote the manuscript. All authors contributed to the final manuscript. The funders had no role in study design, data collection and analysis, decision to publish, or preparation of the manuscript.

## Additional information

**Competing interests:** The authors declare no competing interests.

