## [Peer Review File · Nature Communications]

Reviewers' comments:

Reviewer #1 (Remarks to the Author):

The manuscript by Mirouze et al is an interesting and potentially important contribution to the transformation literature. The identity of the initial DNA binding step has been something of a mystery and this study sheds light, is novel and will likely lead to future interesting work. Here are a few comments that might improve the manuscript.

1. In several places it is suggested that modified WTA is the molecule that directly binds to DNA. I believe that the authors have convincingly demonstrated that WTA, decorated with some unknown modification, possibly a glycosidic addition, is involved in binding, but there is no evidence that it interacts directly with DNA. I think the text should state this explicitly and the several sentences that suggest direct binding should be qualified.
2. Ref. 16 states that when tuaH is inactivated, uronate is not incorporated into teichuronic acid. The present manuscript suggests that TuaH modifies WTA. If the modification involves the addition of uronic acid, this would not affect the anionic nature of teichoic acid and would not help DNA bind, as suggested for the modification. Of course uronic acid is simply an oxidized glucose and perhaps TuaH can handle glucose or a glucose derivative. But the addition of glucose would also not modify the negative charge on WTA. Some discussion of these issues would be helpful.
3. The role of ComGA in relation to WTA is mysterious. ComGA is needed for binding of DNA and also reportedly for construction of a polymer containing ComGC. How are these related to WTA. This issue deserved better treatment in the discussion with some comment, at least acknowledging that this is an area of interest and puzzlement. The speculation (lines 370-371) that ComGA might interact with TuaH does not ascribe a role for the so-called pseudopilus.
4. Fig. 1 shows a severe growth effect of tunicamycin addition (5 ug/ml), clearly visible within the first hour of growth. But in Fig. Sup2a there is no effect for about 2 hours and only a slight effect for the next three hours. How reproducible is the effect on growth?
5. In Fig. S2b it is shown that tunicamycin has no effect on the % cells expressing comK. But the growth in the second step of the two-step procedure is largely due to the non-comK-expressing cells because competent cells are not growing and dividing. If this growth is inhibited, as shown in Fig.1, why is the % of expressing cells almost exactly the same with or without tunicamycin?
6. The lowest concentration of tunicamycin is established in the medium used for competence. This concentration is then used to show complementation of growth and morphology in LB (Fig. 2a and Fig.3Sb). Then complementation of transformation is shown. This is very nice indeed. But it would be even better if the complementation of growth/morphology were shown in SPII.
7. In Fig. 3S the Δ tagO mutant seems to not grow at all. How is this strain manipulated?
8. Strikingly, Fig. 1D shows that targocil causes a LOSS of transformability. Is this accompanied by lysis of competent cells? Or what? Comment?
9. It is interesting that there is enhanced staining of competent cells by ComA. This is interpreted as meaning that there is more WTA (line 231). Are the tag genes overexpressed in these cells? Perhaps instead there is a different degree of accessibility to ComA?
10. Line 260: "virtually" ? How many cells with no dots?
11. Are there ComK-boxes upstream of tuaH? Any explanation for the results from micro-array. Ogura and Hamoen have also published transcriptional profiles of ComK. Do they also show increased tuaH expression? If so, this is good support. If not it creates a doubt. In either case they should be referenced.
12. It is important to verify micro-array data because they are often unreliable for a given gene. If no verification is available, the over expression statements should be qualified.
13. Line 396: At least one prior paper from the Dubnau lab showed that cellular elongation is retarded in competent cells. It should also be referenced.
14. Here are a few English corrections:
 - a. Everywhere: not two-steps but two-step
 - b. line 22: responsible for
 - c. line 77: taken up

- d. line 126: appearance
- e. line 280: upstream of tuaF
- f. line 281: in addition to the
- g. line 302: upstream of
- h. line 313L associated with

Reviewer #2 (Remarks to the Author):

The work by Mirouze et al. addresses a critical question in the field of natural transformation, which is the identity of the cellular factor(s) responsible for initial binding of DNA to the cell. The authors provide a series of generally well-executed and properly controlled experiments which implicate wall teichoic acids in this binding event. The authors correlate the responses to antibiotics with different effects on the cell wall with defects in transformation. Based on the cellular targets of those antibiotics, the authors then test the effects more directly by examining transformation efficiencies of genetic knockouts in those respective cell wall biosynthesis pathways. Finally, using fluorescence microscopy the authors correlate cell surface DNA interactions with the presence of both competence proteins as well as with WTA. They summarize their findings with the proposal that competence-specific WTAs are produced and modified and that these structures are the critical missing link for DNA binding.

I would like to see the following comments addressed:

1. One of the major points made in the abstract, is that “WTA specifically produced and modified during competence” (p1, line 25) are important for this binding event. The authors have shown a link between WTA biosynthetic genes and DNA interaction at the cell surface. They have not, however, directly demonstrated the WTA is specifically newly synthesized nor that there are modifications that directly mediate the DNA interaction. Inferring these is reasonable, but requires additional direct evidence. WTA extraction and analysis would be needed to make this statement (see for example Meredith, Swoboda, Walker, 2008 JBact). For example, it could be that proteins responsible for the DNA binding are unstable or inaccessible to the surface in these cell wall mutant backgrounds. Such a model would produce the same results as the authors show. In either case, it is clear that the state of the cell wall, and specifically the WTA, is important for the DNA binding.
2. To be consistent with the authors’ proposal that there are competence specific WTAs and that tunicamycin has these two effects, 1. preventing the WTA acid synthesis in competent cells and in turn preventing DNA binding, and 2. Preventing synthesis of vegetative WTA, means that there must be a competence specific WTA identifiable. Further, it would imply that timing of the tunicamycin should be a critical component in the action of the drug in the presented experiments. An alternate (or complementary) approach to profiling the WTAs would be to do a time series of drug addition. The current experiments add drug at a singular time point, T0, or the transition from SPI to SPII. At the point where cells are diluted from SPI to SPII the cells are reentering growth phase such that it seems that the results are produced from a mixture of inhibition of the presumed two distinct targets that the authors indirectly imply in their model.

3. The proximity of ComGA foci to ConA intensity peaks (0.68 μm) is used to argue that the WTA is enriched near competence proteins and this helps in cell delivery of DNA. This argument could be strengthened if an appropriate statistical analysis were included to distinguish what a random distribution would produce in terms of proximities for the same data set. For example, if one looks only at cell #1 in Figure 3C, there are three ComGA foci, and there is one ConA peak. If one had a population of cell with three ComGA peaks that were all this same length, if the ConA peak were randomly distributed along the lengths of those cells, what would the average distance to the closest ComGA peak be. Now, this same type of analysis can be done on the current data set for the full distribution of cell lengths, numbers of ComGA foci and ConA peaks, to see whether the proximity reported is truly shorter than what would be expected by chance. Looking at the three example cells given, it does not appear that random distribution would produce longer spacings. To be able to make the conclusions that the authors make from these data, the modeling and statistics should show that the data are better than chance. The choice of what constitutes peaks for the ConA labeling makes it look as though short distances would have been reportable for Cell #3 as well, even if there has been ComGA foci in the middle.

Additional points that could use clarification:

1. p3 line 8/Figure 3/p26 line 591/Supp table 1/p34 line 751 and other places

There are two gfp reporters mentioned in the text and it is unclear whether in fact two in use. In some cases the reporter is listed as a promoter fusion PcomK-gfp, and in other cases as a translational fusion PcomK-comK-gfp. If it is the former which is being used in all cases, then the protein needs to be properly referenced as "GFP" and not "ComK-GFP". If it is the latter, then a few legends are incorrectly labeled with the genotype. If both of these are actually being used, then why the switch?

2. p30 line 648

If not identical to previous method, then please report differences. "similar" implies changes have been made to the protocol.

3. Fig 3a

What do the error bars represent here? These are reported as analysis of a % of cells binding DNA and the N of cells is given as >2000 or > 400. So were there replicates of each N done and these are averaged with SE shown?

4. Fig 2

What do the error bars represent here? Are these biological or technical replicates? How many? 4. Fig 1d - Legend does not indicate induction status – IPTG added and when? - It is interesting that this genetic background seems to produce improved growth at the longer time points for the untreated cells, yet transformability is unchanged. Is this dependent on the orthogonal tarGH? Presumably it is? Any ideas what is going on here? It seems a potentially useful insight to the mechanism or modification.

5. Supp Fig 7

The authors state that washing removed DNA and that fl-ConA inhibited DNA binding, so could the binding shown here merely be due to selectively retaining only DNA in non-ConA labeled regions, i.e. regions with higher binding affinity, rather than of a tunicamycin phenotype?

Reviewer #3 (Remarks to the Author):

Despite that transformation is one of the important ways of horizontal gene transfers between bacteria, the initial binding of foreign DNA to bacterial cell surface structures has been hardly investigated. This manuscript aims to clarify which bacterial surface structures the transforming DNA will firstly bind to before uptake by the bacterial cell.

In this manuscript, the authors confirmed that while beta-lactam and fosfomycin have no effect, tunicamycin and targocil, two antibiotics targeting wall teichoic acids (WTA) biogenesis pathways, can drastically reduce transformation efficiency of *Bacillus subtilis* wild type strain 168 (Fig1) . It was previously shown that low concentration Tunicamycin specifically targets tagO, the enzyme catalyzed the first step of WTA biosynthesis in *S aureus*. The authors showed that tunicamycin strongly inhibit bacterial growth and does not have any effect on the expression of competence gene (Fig S2) . Importantly, they also demonstrated that overexpression of tagO in either wild type or tagO deletion mutant can overcome the transformation inhibition effect of tunicamycin , suggesting that transformation of bacillus subtilis strain 168 requires WTA (fig 3) .

As WTAs are covalently linked to the peptidoglycan and localized at the outmost layer of the bacterial cell surface , the author hypothesized that the transforming DNA need firstly bind to WTA before uptake by bacterial cells and inhibition of WTA biogenesis will result in the loss of binding sites for foreign DNA and eventually leading to reduced transformation efficiency .

To prove this hypothesis , the authors carried out florescence microscopy studies aiming to localize foreign DNA , glucosylated WTA and competence maker proteins in the competent cells. Using comK-gfp as the marker for competent cells, they observed that transforming DNA binds to the polar regions of the competent cell and the percentage of DNA binding competent cells drastically reduced either when the culture was treated with 5ug/ml tunicamycin (figure 3A: bar 3 vs bar 4) or when tuaH gene is deleted (figure 3A : bar 3 vs bar 7 (from left to right) . Notably tuaH encodes a putative glycosyltransferase gene whose expression is induced during competence and it is localized directly upstream of tagO gene (Fig S9) . Additionally their preliminary studies of tua mutants suggest TuaH activity in transformation is independent of other teichuronic acid genes (table 1) . Based on all these data, they proposed that TuaH can specifically glycosylate WTA during competence and promote DNA binding to WTA .

Although this is an interesting model, the authors do not have enough direct evidence to support this model. Below are my major concerns:

A. The structural difference of vegetative WTA (vWTA) and competence WTA (cWTA) ?

It is interesting that the authors indicated in their model the structural difference between vegetative WTA and competence WTA (fig 5) . However the author did not explain what the structural differences are between vWTA and cWTA. Do they have different alanine modification or glycosylation pattern?

In *Bacillus subtilis* strain 168 , both WTA and lipoteichoic acids (LTA) are composed of repetitive glycerolphosphate (GroP) . while C2 –OH of GroP in 168 WTA can be modified either by D-Alanine or alpha- Glc , the C2-OH of GroP could be substituted either with D-Ala or alpha –GlcNAc . The glycosylation of WTA and LTA are respectively catalyzed by TagE and YfhO . The alanine modification of both WTA and LTA are dependent on the activities of the Dlt proteins encoded by the operon dltABCD

It would be beneficial to check the expression of tagE and dltABCD during vegetative growth and competency development

B. In Figure 4b , The authors used fl-conA as probe to detect glucosylated WTA. However to ensure the conA signal is specific to glucose residues on WTA and exclude any noise signal caused by conA binding to peptidoglycan or GlcNAc residues on LTA , they should consider to include the following knockout mutants in the microscopy studies:

- 1) tagE mutant: no WTA glc but possibly increased D-Ala
- 2) dltA mutant: no D-Ala on LTA and WTA
- 3) double mutant deficient in tagE and dltA : no D-ala on LTA and WTA and no glc on WTA.
- 4) yfhO or CSbH mutant : no GlcNAc on LTA but possibly increase d-Ala on LTA
- 5) Double mutant deficient in yfhO and dltA : no D-ala on LTA and WTA and no GlcNAc on LTA.
- 6) tagO and tagA mutant : no WTA
- 7) ItaS conditional knockout: no LTA

C: WTA structural variation during competence

in both Fig5A and 5B , what are the concentrations of tunicamycin used in this study ? 5ug/ml or 0.31 ug/ml ? have the author checked if Tunicamycin at the concentration of 0.31 ug/ml can completely block the WTA synthesis ?

Previously it was shown that tunicamycin at the concentration of 0,5 ug/ml was able to completely block WAT biosynthesis in *S aureus* (Campbell J 2011) ., it is very likely that the WTA biogenesis in the wild type 168 cells is completely blocked and there will not be any type of WTA on the cell surface when the cells were treated with 5ug/ml tunicamycin .

D: DNA binding studies (Figure 3A , 3D , Fig 7)

Would this DNA staining be inhibited by high salt concentration?

In addition to WTA and LTA , there are other polymers present in the cell envelope, for example the bacillus minor WTA and teichuronic acids .

As WTA would completely be blocked when cells were treated with 5ug/ml tunicamycin . would it be that this increase of the percentage of DNA binding non-competent cells (fig 3A bar2 vs bar 1) is caused by the exposure of other DNA binding sites in the cell envelope, for example GlcNAc modified Lipoteichoic acids , 168 minor WTA or teichuronic acids ? It would be beneficial to repeat the DNA binding studies (also fig 7) with the same mutant panel suggested above .

E: No direct proof supporting that TuaH is a cWTA glycosyltransferase . (line 348-356)

The author hypothesized that TuaH is a glycosyltransferase responsible for synthesis of competence WTA based on the following facts: 1) tuaH encodes a putative glycotransferase and localized directly upstream of tagO gene 2) its expression is induced during competence development 3) genetic analysis showed that the involvement of tuaH in transformation is independent other tua genes .

However the role of tuaH in bacterial competence seems very different from that of WTA . As shown in table 1, tagO deletion and tuaH deletion lead to 1000 fold and 80 fold reduction of transformation efficiency, respectively . This suggest that TagO activity and WTA are essential for transformation while TuaH activity can facilitate the transformation.

To confirm the role of tuaH in transformation, it would be helpful to create a tauH mutant with genetic background different from strain 168, for example , strain W23 which produces a different type of WTA, the polyribitolphosphate type WTA modified with beta-Glc

To prove TuaH is a WTA glycosyltransferase , it is necessary to test its in vitro enzyme activity and specificity towards non-glycosylated WTA , LTA and maybe other secondary cell wall polymers .

In addition, , it would be helpful to investigate if overexpression of tagE or yfhO in wild type 168 lead to increase or decrease in transformation efficiency , if tagE or yfhO can complement tuaH mutation and if they can inhibit TuaH activity .

I have found a few minors:

- 1) page 7 , line 145, replace 'important' with 'severe '
- 2) page 7, line 157, confirmed that Tunicamycin , a WTA targeting antibiotic, inhibits transformation
- 3) Page 8, line 184 , 64% displayed one DNA molecule at their surface

Not sure what one DNA molecule means here .

4) Line 204, and line 358 , conA detect glucose residues , not ' GlcNAc'

5) Line 454: should read tuaH , not 'tago'

Reviewer #1 (Remarks to the Author):

The manuscript by Mirouze et al is an interesting and potentially important contribution to the transformation literature. The identity of the initial DNA binding step has been something of a mystery and this study sheds light, is novel and will likely lead to future interesting work. Here are a few comments that might improve the manuscript.

1. In several places it is suggested that modified WTA is the molecule that directly binds to DNA. I believe that the authors have convincingly demonstrated that WTA, decorated with some unknown modification, possibly a glycosidic addition, is involved in binding, but there is no evidence that it interacts directly with DNA. I think the text should state this explicitly and the several sentences that suggest direct binding should be qualified.

We agree with Reviewer #1 that we have no evidence for a direct binding. This is why in the original manuscript we tried to use the terms 'enable' and 'promote' which do not necessarily imply a 'direct' binding. To remove any possible ambiguity, we now explicitly mention in several places in the text (lines 25, 110, 200, 344, 374 and 431) that DNA binding to WTA could be direct or indirect.

2. Ref. 16 states that when tuaH is inactivated, uronate is not incorporated into teichuronic acid. The present manuscript suggests that TuaH modifies WTA. If the modification involves the addition of uronic acid, this would not affect the anionic nature of teichoic acid and would not help DNA bind, as suggested for the modification. Of course uronic acid is simply an oxidized glucose and perhaps TuaH can handle glucose or a glucose derivative. But the addition of glucose would also not modify the negative charge on WTA. Some discussion of these issues would be helpful.

In ref. 16 the authors did not conclusively show that TuaH incorporates uronate in teichuronic acids. As shown in Table 2, they measured the amount of uronate in the walls of different *B. subtilis* tua mutants. Even though uronate was not incorporated in a tuaH mutant, it was also the case in tuaA, B, D and G mutants (ref. 16, Table 2). In fact, any protein involved before uronate incorporation would lead to a decrease of uronic acid in *B. subtilis* walls.

In addition, the authors state that the tua operon contains three potential glycosyl-transferases:

'Paradoxically, up to two of the three remaining Tua proteins, i.e. TuaC, G and H, all possibly endowed with a transglycosylase activity, would be in excess of enzymatic activities required for the synthesis of poly(glucuronyl N-acetylgalactosamine) (Fig. 4). Therefore, considering the number of genes shown to be involved in teichuronic acid synthesis, it is likely that either the polymer or its mode of synthesis or both are more complex than presently accepted.'

In conclusion, no clear function has been associated to TuaH. Thus we believe that it is legitimate to propose that this putative glycosyl-transferase could modify WTA by transferring other sugars during competence.

3. The role of ComGA in relation to WTA is mysterious. ComGA is needed for binding of DNA and also reportedly for construction of a polymer containing ComGC. How are these related to WTA. This issue deserved better treatment in the discussion with some comment, at least acknowledging that this is an area of interest and puzzlement. The speculation (lines 370-371) that ComGA might interact with TuaH does not ascribe a role for the so-called pseudopilus.

The roles/functions of ComGA regarding DNA binding/transport remain indeed mysterious. In the discussion, we propose that ComGA could localize TuaH or other Tag proteins in order to promote

WTA localization/modification at the vicinity of the transformation apparatus. This would be a plausible explanation to how a cytosolic protein, ComGA, could be essential for DNA binding at the surface of competent cells but remains speculative on the basis of the current available data.

The fact that ComGA is essential for both DNA binding and DNA transport (through the pseudopilus synthesis) is indeed of interest. We followed reviewer#1 advice and added few sentences in the discussion (line 395).

4. Fig. 1 shows a severe growth effect of tunicamycin addition (5 ug/ml), clearly visible within the first hour of growth. But in Fig. Sup2a there is no effect for about 2 hours and only a slight effect for the next three hours. How reproducible is the effect on growth?

The curves shown in Fig. 1 and Fig. S2 are not to be directly compared since they do not represent the same thing. In Fig. 1, we show the evolution of viability (CFU/ml) and of transformation efficiency in the absence and in the presence of different antibiotics (including tunicamycin) after dilution in SPII. In Fig. S2a, growth is followed by optical density measurements.

We also measured the evolution of OD in the experiment shown in Fig. 1 and the curve is comparable to the growth curve shown in Fig. S2a (figure shown below). This is true for all the antibiotics used here. Therefore, the growth effect is very reproducible.

One explanation could be that non-competent cells try to grow in the presence of tunicamycin. As shown in Fig 2a, tunicamycin greatly affects cell wall synthesis/organization leading to bugged non-competent cells which increase their volume. This abnormal morphology could then participate to the increase in OD observed. However, these cells are no longer viable, because of their impaired cell wall organization, leading to a constant and low viability.

5. In Fig. S2b it is shown that tunicamycin has no effect on the % cells expressing comK. But the growth in the second step of the two-step procedure is largely due to the non-comK-expressing cells because competent cells are not growing and dividing. If this growth is inhibited, as shown in Fig.1, why is the % of expressing cells almost exactly the same with or without tunicamycin?

At the time of dilution, there are very few transformants (see Supplementary Fig 1) which mainly appear during the first hour after dilution. During this hour, non-competent cells only divide once on average (see Fig1, viability panels). As a consequence, addition of tunicamycin does not affect that much the total number of cells (only by a factor 2) and to keep the percentage constant the number of competence expressing cells should only decrease by the same factor (i.e. factor 2).

In addition, the competent cells observed under the microscope never displayed morphological defects due to the addition of tunicamycin. This probably indicates that the decision to induce competence is taken relatively early in SPII, before the cells are affected by the antibiotic and therefore at times where the total number of non-competent cells is not different with or without antibiotic.

6. The lowest concentration of tunicamycin is established in the medium used for competence. This concentration is then used to show complementation of growth and morphology in LB (Fig. 2a and Fig.3Sb). Then complementation of transformation is shown. This is very nice indeed. But it would be even better if the complementation of growth/morphology were shown in SPII.

We originally wanted to show that the growth and morphology complementation occurred in different growth media. Furthermore, the *tagO* morphology phenotype reported in the literature was mostly produced in LB and we wanted to show that it was reproducible in our hands.

We have nevertheless followed the advice of the referee and replaced the growth and morphology complementation experiments in LB by the medium we used to induce competence (updated Fig 2a and Supplementary Fig. 3). We kept the phenotypic complementation in LB in Supplementary Fig. 3 as a reference.

7. In Fig. 3S the $\Delta tagO$ mutant seems to not grow at all. How is this strain manipulated?

We found that, like many cell wall-related mutants, the morphological and growth phenotypes of the $\Delta tagO$ mutant can be complemented by addition of high Mg^{2+} concentrations in the medium. Addition of high Mg^{2+} allows us to manipulate the mutant before the experiment (on plate or liquid over-night cultures). We, of course, removed the magnesium during the experiment in LB to reveal the $\Delta tagO$ phenotypes.

Note that, consistently, the *tagO* mutant strain grows better in SPI than previously in LB as SPI contains 6mM Mg^{2+} (new and old version, respectively, of Fig S3).

8. Strikingly, Fig. 1D shows that targocil causes a LOSS of transformability. Is this accompanied by lysis of competent cells? Or what? Comment?

Great point. We thank reviewer #1 to spot this loss of transformants in Fig.1. We revisited the experiments and realized that we had made a mistake in the original concentration of targocil used, which was too high. This probably induced the lysis of competent cells as suggested by reviewer#1. We repeated the experiment (x 3 times) using the right concentration of targocil (275 ug/ml, as indicated in Fig. 1 legend) and obtained a profile comparable to that of tunicamycin.

We have now updated Fig 1D accordingly. We apologize for this mistake and thank again reviewer #1 for spotting this.

9. It is interesting that there is enhanced staining of competent cells by ComA. This is interpreted as meaning that there is more WTA (line 231). Are the tag genes overexpressed in these cells? Perhaps instead there is a different degree of accessibility to ConA?

tag genes were not found to be expressed in the transcriptomic study by (Berka *et al*, 2002). We now acknowledge in the text (line 244) that we cannot exclude that accessibility to ConA is modified.

However, we found that WTA synthesis (TagO) and export (TagGH) are required for genetic transformation so the most plausible explanation is that more WTAs are produced and that the increased staining is coming from more WTAs in the cell surface.

10. Line 260: “virtually” ? How many cells with no dots?

In the text, we wrote: ‘We grew $\Delta tagO$ mutant cells to exponential phase in rich LB medium and observed that these cells frequently bound exogenous DNA (Supplementary Fig. 8a), while virtually no DNA binding was observed on the surface of wild-type cells growing in the same conditions (Supplementary Fig. 8b).’

We said ‘virtually’ has 2% of the cells displayed DNA at their surface, a number comparable to non-competent cells in SPII medium. We have rephrased the sentence line 272 to report the exact percentage: ‘while only 2% of wild-type cells growing in the same conditions bound DNA (Supplementary Fig. 8b).’

11. Are there ComK-boxes upstream of *tuaH*? Any explanation for the results from micro-array. Ogura and Hamoen have also published transcriptional profiles of ComK. Do they also show increased *tuaH* expression? If so, this is good support. If not it creates a doubt. In either case they should be referenced.

We could not identify canonical ComK-boxes in front of *tuaFGH*. However, if ComK-boxes were present and based on the ‘low’ overexpression of these genes, these boxes would probably be degenerated. Besides, activation of *tuAFGH* expression does not have to be directly controlled by ComK. An intermediate regulator could be involved, implying that no ComK boxes would then be present.

tuaH is not found in (Ogura *et al*, 2002) and (Hamoen *et al*, 2002) papers. There is not an 100% fit between these studies, especially if we consider genes for which the overexpression factor is low. Please see our answer to the next point for our new experimental proof that *tuaFGH* are overexpressed during competence.

12. It is important to verify micro-array data because they are often unreliable for a given gene. If no verification is available, the over expression statements should be qualified.

We followed reviewer #1 advice and verified *tuaFGH* overexpression during competence development using the luciferase from *Photinus pyralis*. To do so, we constructed transcriptional fusions between 1 kbp fragments present upstream of *tuaA*, *tuaF* and *tuaH* and the luciferase gene. These fusions were inserted at the native *tua* loci by single cross-over, in wild type cells and in cells harboring a multicopy plasmid containing the *comS* gene (mccomS). The mcomS construct completely bypass the normal competence regulatory pathways allowing ComK, the competence master regulator to be stable during exponential growth. Therefore, the mcomS construct allows us to increase the percentage of competent cells in the culture to nearly 60%, a strategy already used in D. Dubnau's lab for the transcriptional analysis of a *mecA* mutant (Berka *et al*, 2002). As shown in the new Supplementary Fig 12, the luciferase experiments confirmed that *tuaH* is specifically expressed during competence.

We now include this result in the manuscript (lines 298-303 and Supplementary Fig 12).

13. Line 396: At least one prior paper from the Dubnau lab showed that cellular elongation is retarded in competent cells. It should also be referenced.

Sorry for this unintentional omission/mistake. Dubnau's lab publication is now properly referenced (Haijema *et al*, 2001).

14. Here are a few English corrections: Done

- a. Everywhere: not two-steps but two-step
- b. line 22: responsible for
- c. line 77: taken up
- d. line 126: appearance
- e. line 280: upstream of *tuaF*
- f. line 281: in addition to the
- g. line 302: upstream of
- h. line 313L associated with

Reviewer #2 (Remarks to the Author):

The work by Mirouze et al. addresses a critical question in the field of natural transformation, which is the identity of the cellular factor(s) responsible for initial binding of DNA to the cell. The authors provide a series of generally well-executed and properly controlled experiments which implicate wall teichoic acids in this binding event. The authors correlate the responses to antibiotics with different effects on the cell wall with defects in transformation. Based on the cellular targets of those antibiotics, the authors then test the effects more directly by examining transformation efficiencies of genetic knockouts in those respective cell wall biosynthesis pathways. Finally, using fluorescence microscopy the authors correlate cell surface DNA interactions with the presence of both competence proteins as well as with WTA. They summarize their findings with the proposal that competence-specific WTAs are produced and modified and that these structures are the critical missing link for DNA binding.

I would like to see the following comments addressed:

1. One of the major points made in the abstract, is that “WTA specifically produced and modified during competence” (p1, line 25) are important for this binding event. The authors have shown a link between WTA biosynthetic genes and DNA interaction at the cell surface. They have not, however, directly demonstrated the WTA is specifically newly synthesized nor that there are modifications that directly mediate the DNA interaction. Inferring these is reasonable, but requires additional direct evidence. WTA extraction and analysis would be needed to make this statement (see for example Meredith, Swoboda, Walker, 2008 JBact). For example, it could be that proteins responsible for the DNA binding are unstable or inaccessible to the surface in these cell wall mutant backgrounds. Such a model would produce the same results as the authors show. In either case, it is clear that the state of the cell wall, and specifically the WTA, is important for the DNA binding.

Reviewer #2 says that we haven't demonstrated that newly synthesized WTA participate to DNA binding. We respectfully disagree. We do show in our manuscript that both active WTA synthesis (TagO activity, see Fig. 2b) and export (TagGH activity, see Fig. 1d) are essential for DNA binding and transformation. The comment of the reviewer nevertheless indicates that this point did not come across clearly in our manuscript. We have therefore performed the alternative experiment proposed by reviewer #2 in his/her next point.

Then, reviewer #2 says that we haven't shown that modifications directly mediate the DNA interaction. We agree with this point. In the manuscript we tried to convey the message that even though WTA are essential for DNA binding during genetic transformation we do not know if this binding is direct or indirect (requiring a third actor). As indicated in our answer to point 1 of reviewer #1, we now explicitly state this at several places in our manuscript.

Finally, we agree that extraction and analysis of WTA would directly show if there are competence-specific modifications. Such experiments have however important limitations. Because of the bistable nature of competence in *B. subtilis* (only 5% of cells in the population enter this developmental process), competence-induced WTA would represent a very small fraction of the total WTA purified. Vegetative WTA cover the entire surface of non-competent cells (i.e. 95% of the culture) and are also present at the surface of competent-cells that were vegetative before becoming competent. Thus, competence-specific WTAs would be masked (diluted) by vegetative WTAs in the biochemical analysis. Setting up the conditions for this analysis would represent an entire project and is therefore beyond the scope of this publication.

2. To be consistent with the authors' proposal that there are competence specific WTAs and that tunicamycin has these two effects, 1. preventing the WTA acid synthesis in competent cells and in turn preventing DNA binding, and 2. Preventing synthesis of vegetative WTA, means that there must be a competence specific WTA identifiable. Further, it would imply that timing of the tunicamycin should be a critical component in the action of the drug in the presented experiments. An alternate (or complementary) approach to profiling the WTAs would be to do a time series of

drug addition. The current experiments add drug at a singular time point, T₀, or the transition from SPI to SPII. At the point where cells are diluted from SPI to SPII the cells are reentering growth phase such that it seems that the results are produced from a mixture of inhibition of the presumed two distinct targets that the authors indirectly imply in their model.

We thank reviewer #2 for this thoughtful suggestion. We believe that this experiment confirms the need to synthesize new WTA during competence, as an alternative experiment to the WTA profiling suggested in point 1. We performed a time series experiment in which we added tunicamycin at different time points after dilution in medium II and monitored the effect of this antibiotic on the apparition of transformants, as suggested. As expected in a model where newly synthesized WTA are essential for genetic transformation, the later we added the tunicamycin the more transformants were obtained.

These results are now presented in the manuscript in the new Supplemental Fig 5. We also mention this new result in the main text line 162.

3. The proximity of ComGA foci to ConA intensity peaks (0.68 μm) is used to argue that the WTA is enriched near competence proteins and this helps in cell delivery of DNA. This argument could be strengthened if an appropriate statistical analysis were included to distinguish what a random distribution would produce in terms of proximities for the same data set. For example, if one looks only at cell #1 in Figure 3C, there are three ComGA foci, and there is one ConA peak. If one had a population of cell with three ComGA peaks that were all this same length, if the ConA peak were randomly distributed along the lengths of those cells, what would the average distance to the closest ComGA peak be. Now, this same type of analysis can be done on the current data set for the full distribution of cell lengths, numbers of ComGA foci and ConA peaks, to see whether the proximity reported is truly shorter than what would be expected by chance. Looking at the three example cells given, it does not appear that random distribution would produce longer spacings. To be able to make the conclusions that the authors make from these data, the modeling and statistics should show that the data are better than chance. The choice of what constitutes peaks for the ConA labeling makes it look as though short distances would have been reportable for Cell #3 as well, even if there has been ComGA foci in the middle.

We appreciate the reviewer's consideration and believe that it was important to strengthen this point in the manuscript. As mentioned in the text, a majority of competent cells only display 1 ComGA focus at one pole or one ComGA focus at both poles (>75%). Considering the average length shown in Fig 3a (3,5 μm) and the average ComGA-WTA distance measured (0,71 μm), a random distribution could not provide such bias. We originally decided to show cells with multiple ComGA foci in Fig 3c in order to provide more localization examples in a limited available Figure space.

To reflect more precisely the reality of ComGA localization distribution among competent cells, we removed the first cell and replaced it by two cells with one or two ComGA polar cluster.

Additional points that could use clarification:

1. p3 line 8/Figure 3/p26 line 591/Supp table 1/p34 line 751 and other places. There are two gfp reporters mentioned in the text and it is unclear whether in fact two in use. In some cases the reporter is listed as a promoter fusion PcomK-gfp, and in other cases as a translational fusions PcomK-comK-gfp. If is the former which is being used in all cases, then the protein needs to be properly referenced as "GFP" and not "ComK-GFP". If it is the later, then a few legends are incorrectly labeled with the genotype. If both of these are actually being used, then why the switch?

There is only one fusion: PcomK-comK-gfp. We are sorry for the confusion. We corrected the text.

2. p30 line 648. If not identical to previous method, then please report differences. "similar" implies changes have been made to the protocol.

We replaced 'similar' by 'identical'.

3. Fig 3a. What do the error bars represent here? These are reported as analysis of a % of cells binding DNA and the N of cells is given as >2000 or > 400. So were there replicated of each N done and these are averaged with SE shown?

Three independent experiments were conducted on three different days, with >2000 or >400 cells measured for each experiment. The percentage of cells binding DNA was calculated for each independent experiment, and the three percentages were then averaged (SE of the average is shown). We now explain how these data were calculated in Fig 3 legend.

4. Fig 2. What do the error bars represent here? Are these biological or technical replicates? How many?

They are biological replicates. The experiments were repeated three times on different days. Fig 2b shows the averaged transformation efficiencies. The error bars represent the SE of the average. This is now mentioned in Fig. 2 legend.

4. Fig 1d.

- Legend does not indicate induction status – IPTG added and when?

This is now mentioned in the Fig 1 legend.

- It is interesting that this genetic background seems to produce improved growth at the longer time points for the untreated cells, yet transformability is unchanged. Is this dependent on the orthogonal tarGH? Presumably it is? Any ideas what is going on here? It seems a potentially useful insight to the mechanism or modification.

We repeated the experiment shown in Fig. 1d to address a question raised by reviewer #1. In the new viability curve, this improved growth does not seem evident. As noted by reviewer #2, this ‘improved growth’ could only be seen with the longer time point.

We agree with reviewer #2 that it would be interesting if expression of the orthogonal *tarGH* improved growth. Please note, however, that the growth observed is only attributed to the non-competent cells. We are not sure if we could extrapolate interesting hypotheses to explain the mechanism of WTA modification in the non-growing competent-cells.

5. Supp Fig 7. The authors state that washing removed DNA and that fl-ConA inhibited DNA binding, so could the binding shown here merely be due to selectively retaining only DNA in non-ConA labeled regions, i.e. regions with higher binding affinity, rather than of a tunicamycin phenotype?

We agree that this result does not reflect a tunicamycin phenotype and rather shows that the accessibility to DNA receptors is revealed in the absence of vegetative WTA.

This result reveals that vegetative cell walls contain DNA receptors that are normally not exposed. In itself, this result is not important for this study. However, it allows us to conclude that vegetative WTA normally inhibit DNA binding at the surface of non-competent cells. More importantly, we also use this last conclusion to propose that in order to explain the different DNA binding affinity of vegetative and competence-induced WTA, they must have a different composition. We hope that this demonstration is clear in the manuscript.

Reviewer #3 (Remarks to the Author):

Despite that transformation is one of the important ways of horizontal gene transfers between bacteria, the initial binding of foreign DNA to bacterial cell surface structures has been hardly investigated. This manuscript aims to clarify which bacterial surface structures the transforming DNA will firstly bind to before uptake by the bacterial cell.

In this manuscript, the authors confirmed that while beta-lactam and fosfomycin have no effect, tunicamycin and targocil, two antibiotics targeting wall teichoic acids (WTA) biogenesis pathways, can drastically reduce transformation efficiency of *Bacillus subtilis* wild type strain 168 (Fig1).

It was previously shown that low concentration Tunicamycin specifically targets tagO, the enzyme catalyzed the first step of WTA biosynthesis in *S aureus*. The authors showed that tunicamycin strongly inhibit bacterial growth and does not have any effect on the expression of competence gene (Fig S2).

Importantly, they also demonstrated that overexpression of tagO in either wild type or tagO deletion mutant can overcome the transformation inhibition effect of tunicamycin, suggesting that transformation of *bacillus subtilis* strain 168 requires WTA (fig. 3).

As WTAs are covalently linked to the peptidoglycan and localized at the outmost layer of the bacterial cell surface , the author hypothesized that the transforming DNA need firstly bind to WTA before uptake by bacterial cells and inhibition of WTA biogenesis will result in the loss of binding sites for foreign DNA and eventually leading to reduced transformation efficiency.

To prove this hypothesis , the authors carried out florescence microscopy studies aiming to localize foreign DNA , glucosylated WTA and competence maker proteins in the competent cells. Using comK-gfp as the marker for competent cells, they observed that transforming DNA binds to the polar regions of the competent cell and the percentage of DNA binding competent cells drastically reduced either when the culture was treated with 5ug/ml tunicamycin (figure 3A: bar 3 vs bar 4) or when tuaH gene is deleted (figure 3A : bar 3 vs bar 7 (from left to right)). Notably tuaH encodes a putative glycosyltransferase gene whose expression is induced during competence and it is localized directly upstream of tagO gene (Fig S9) . Additionally their preliminary studies of tua mutants suggest TuaH activity in transformation is independent of other teichuronic acid genes (table 1). Based on all these data, they proposed that TuaH can specifically glycosylate WTA during competence and promote DNA binding to WTA .

Although this is an interesting model, the authors do not have enough direct evidence to support this model. Below are my major concerns:

A. The structural difference of vegetative WTA (vWTA) and competence WTA (cWTA) ?

It is interesting that the authors indicated in their model the structural difference between vegetative WTA and competence WTA (fig 5) . However the author did not explain what the structural differences are between vWTA and cWTA. Do they have different alanine modification or glycosylation pattern?

In *Bacillus subtilis* strain 168 , both WTA and lipoteichoic acids (LTA) are composed of repetitive glycerolphosphate (GroP) . while C2 -OH of GroP in 168 WTA can be modified either by D-Alanine or alpha- Glc , the C2-OH of GroP could be substituted either with D-Ala or alpha - GlcNAc . The glycosylation of WTA and LTA are respectively catalyzed by TagE and YfhO . The alanine modification of both WTA and LTA are dependent on the activities of the Dlt proteins encoded by the operon dltABCD. It would be beneficial to check the expression of tagE and dltABCD during vegetative growth and competency development

Based on the following lines of evidence, we hypothesize that vWTA and cWTA should have structural difference(s) to explain their different ability to prevent or promote, respectively, DNA binding:

- 1/ non-competent cells do not bind DNA while they are covered with vWTA.
- 2/ non-competent cells, treated with tunicamycin, can bind DNA where vWTA are missing
- 3/ competent cells do bind DNA where cWTA accumulate
- 4/ a putative glycosyl-transferase, *tuaH*, is important for DNA binding at the surface of competent cells.

Taken together, these results strongly suggest that vWTA and cWTA have different properties. We did not perform alanine modification or glycosylation patterns on cWTA but, instead, our answer to reviewer #3 point D (below) clearly shows that among the known or putative actors decorating WTA, TuaH is our best candidate to explain the ability of cWTA to bind DNA.

Interestingly, the expression of both *tagE* and *dltABCD* was found to be slightly inhibited during competence development in a recent study in *B. subtilis* (Nicolas *et al*, 2012). The authors used tilling microarrays to characterize the RNA content of cells grown in diverse conditions. Importantly, to monitor the development of competence, the authors used the same two-step protocol used in our study. Ultimately, the fact that genes encoding known enzymes modifying WTA were found down-regulated while *tuaH*, a gene associated to phenotypes during genetic transformation was found up-regulated is in agreement with our model. One could hypothesize that decreasing the amount of WTA glycosylation (*tagE*) and alanilation (*dltABCD*) would be the best option to favor other modifications (for example catalyzed by TuaH).

B. In Figure 4b , The authors used fl-conA as probe to detect glucosylated WTA. However to ensure the conA signal is specific to glucose residues on WTA and exclude any noise signal caused by conA binding to peptidoglycan or GlcNAc residues on LTA , they should consider to include the following knockout mutants in the microscopy studies:

- 1) **tagE mutant: no WTA glc but possibly increased D-Ala**
- 2) **dltA mutant: no D-Ala on LTA and WTA**
- 3) **double mutant deficient in tagE and dltA : no D-ala on LTA and WTA and no glc on WTA.**
- 4) **yfhO or CSbH mutant : no GlcNAc on LTA but possibly increase d-Ala on LTA**
- 5) **Double mutant deficient in yfho and dltA : no D-ala on LTA and WTA and no GlcNAc on LTA.**
- 6) **tagO and tagA mutant : no WTA**
- 7) **ltaS conditional knockout: no LTA**

Thank you. We followed reviewer #3 advice and decided to confirm that fl-ConA really only labels glucose residues on WTA. As suggested by reviewer #3, we first tested a *tagE* mutant, in which, no glucose decorates WTA. In this mutant, we lost all fl-ConA fluorescence at the surface of both non-competent and competent cells. This result indicates that fl-ConA labels specifically glucosylated WTA and thus that there is no noisy signal from PG or LTA. Our result is in full agreement with recently published data by the Garner's lab in (Hussain *et al*, 2018). In their Fig 6, Fig supplement 1, panel A, the authors clearly show that in a *B. subtilis* PY79 background, a *tagE* mutant is susceptible to ConA binding revealing that ConA specifically binds glucose residues decorating WTA.

In addition, the fact that the fl-ConA labeling of competent cells is also completely lost in a *tagE* mutant reveals that the newly synthesized competence-specific WTA are also glucosylated.

These results have been added in the manuscript line 211 and new supplementary Fig. 8.

We also tested the other mutants proposed by reviewer #3 (i.e. *dltA*, *dltA/tagO*, *yfhO* and *ltaS*). As expected, none of these mutants or combination of mutants disrupted the ConA pattern on both competent and non-competent cells. Since all the fl-ConA was lost in a *tagE* mutant background, these results became irrelevant and we did not include them in the manuscript.

C. WTA structural variation during competence

in both Fig5A and 5B , what are the concentrations of tunicamycin used in this study ? 5ug/ml or 0.31 ug/ml ? have the author checked if Tunicamycin at the concentration of 0.31 ug/ml can completely block the WTA synthesis ?

Previously it was shown that tunicamycin at the concentration of 0,5 ug/ml was able to completely block WAT biosynthesis in S aureus (Campbell J 2011) ., it is very likely that the WTA biogenesis in the wild type 168 cells is completely blocked and there will not be any type of WTA on the cell surface when the cells were treated with 5ug/ml tunicamycin.

We are sorry but Fig. 5 in our manuscript is our model. We tried to identify the Fig. that reviewer #3 is referring to in this comment without success. The low tunicamycin concentration used (0,31ug/ml)

shows that TagO activity is the target to inhibit WTA synthesis (Fig 2b). As TagO catalyzes an essential step in this pathway, we believe that WTA synthesis is also completely blocked.

D: DNA binding studies (Figure 3A , 3D , Fig 7)

Would this DNA staining be inhibited by high salt concentration? In addition to WTA and LTA , there are other polymers present in the cell envelope, for example the bacillus minor WTA and teichuronic acids.

As WTA would completely be blocked when cells were treated with 5ug/ml tunicamycin . would it be that this increase of the percentage of DNA binding non-competent cells (fig 3A bar2 vs bar 1) is caused by the exposure of other DNA binding sites in the cell envelope, for example GlcNAc modified Lipoteichoic acids , 168 minor WTA or teichuronic acids ? It would be beneficial to repeat the DNA binding studies (also fig 7) with the same mutant panel suggested above.

The effect of salt concentration on DNA binding has been extensively studied in both *B. subtilis* and *S. pneumoniae* (the other main Gram-positive model bacterium). It was particularly shown that low salt increased the amount of DNA bound by competent cells (Seto & Tomasz, 1975) probably revealing the anionic nature of this superficial binding.

We agree with reviewer #3 that LTA or minor WTA could be involved in DNA binding at the surface of non-competent cells. It is unlikely that TUA would be involved as the *tua* operon is not expressed under these conditions (see Supplementary Fig 12). However, although it could be interesting to understand this second, normally hidden in the cell wall, DNA binding site, investigating DNA binding at the surface of non-competent cells, treated with tunicamycin, is not the purpose of this paper.

Finally, inspired by comment B of reviewer #3, we decided to test if deleting some of the genes involved in the WTA modification would prevent, or not, DNA binding at the surface of competent cells. Interestingly, *tagE* and *dltA* mutants were not affected for their ability to bind DNA (see Supplementary Fig 13). We also completed this study by calculating the transformation efficiency of these mutants. Corroborating our previous result, all these mutants displayed wild-type transformation efficiencies (see Table 1). These experiments clearly show that no known enzyme decorating WTA is able to make cWTA proficient for DNA binding and that TuaH is the best candidate so far (see answer to reviewer #2 point E).

We now present these results line 334, Supplementary Fig 13 and Table 1.

E: No direct proof supporting that TuaH is a cWTA glycosyltransferase. (line 348-356)

The author hypothesized that TuaH is a glycosyltransferase responsible for synthesis of competence WTA based on the following facts: 1) tuaH encodes a putative glycosyltransferase and localized directly upstream of tagO gene 2) its expression is induced during competence development 3) genetic analysis showed that the involvement of tuaH in transformation is independent other tua genes.

However the role of tuaH in bacterial competence seems very different from that of WTA . As shown in table 1, tagO deletion and tuaH deletion lead to 1000 fold and 80 fold reduction of transformation efficiency, respectively . This suggest that TagO activity and WTA are essential for transformation while TuaH activity can facilitate the transformation.

We agree that there is a difference in the transformation efficiencies calculated for tunicamycin treated and *tuaH* mutant cells. A 80-fold decrease still means that almost 99% of the transformants are lost and thus that *tuaH* is a key gene for genetic transformation. As a comparison, deletion of *dprA*, a gene considered as essential and central for the transforming DNA RecA-dependent homologous recombination, displays a similar decrease in transformation in *B. subtilis* (Yadav *et al*, 2014).

However, we agree with the terminology used by reviewer #3 when (s)he says that WTA are essential while TuaH activity facilitates transformation. We can explain it as follows: our model proposes that DNA first binds to modified competence-induced WTA and is then transferred to the transformation apparatus to be transported across the cell wall. One could hypothesize that in the absence of this optimized initial binding sites (i.e. modified competence-induced WTA) some DNA fragments could directly access the transformation apparatus with a much lower probability (i.e. 1%). Hence the fact that TuaH and the associated modification facilitate genetic transformation. On the contrary, when cells are

treated with tunicamycin both DNA binding and transport could be affected. Indeed, both the initial DNA binding sites (i.e. modified competence-induced WTA) and the transformation apparatus could be absent. As WTA are essential to organize cell wall synthesis, one could say that the transformation apparatus construction and proper localization requires intact cell walls. In that case, affecting WTA synthesis would have a dual effect, inhibiting both DNA binding and transport and leading to a more important decrease in the transformation efficiency (i.e. 1000-fold).

To confirm the role of *tuaH* in transformation, it would be helpful to create a *tuaH* mutant with genetic background different from strain 168, for example, strain W23 which produces a different type of WTA, the polyribitolphosphate type WTA modified with beta-Glc.

We tried to construct such mutant in a W23 strain (from the *Bacillus subtilis* stock center, strain TU-B-10T) without success. We verified that the *tuaH* gene was present in this genetic background (it is the case) and that W23 is naturally transformable (which it is, even though at lower efficiencies than 168). We made several attempts to transform the W23 strain using chromosomal DNA of NC241, without success.

To prove *TuaH* is a WTA glycosyltransferase, it is necessary to test its *in vitro* enzyme activity and specificity towards non-glycosylated WTA, LTA and maybe other secondary cell wall polymers.

We agree with reviewer #3 that purifying and testing *in vitro* the activity of *TuaH* would be the perfect way to confirm that it is a WTA glycosyltransferase. This will be addressed by future work in the lab and is beyond the scope of this publication.

In addition, it would be helpful to investigate if overexpression of *tagE* or *yfhO* in wild type 168 lead to increase or decrease in transformation efficiency, if *tagE* or *yfhO* can complement *tuaH* mutation and if they can inhibit *TuaH* activity.

Again, we thank reviewer #3 for this important experiment that we will plan to do in the continuity of this project.

I have found a few minors: Done

- 1) page 7, line 145, replace 'important' with 'severe'
- 2) page 7, line 157, confirmed that Tunicamycin, a WTA targeting antibiotic, inhibits transformation
- 3) Page 8, line 184, 64% displayed one DNA molecule at their surface
Not sure what one DNA molecule means here.
- 4) Line 204, and line 358, *conA* detect glucose residues, not 'GlcNAc'
- 5) Line 454: should read *tuaH*, not 'tago'

References:

- Berka RM, Hahn J, Albano M, Draskovic I, Persuh M, Cui X, Sloma A, Widner W & Dubnau D (2002) Microarray analysis of the *Bacillus subtilis* K-state: genome-wide expression changes dependent on ComK. *Mol Microbiol* **43**: 1331–1345 Available at: <http://www.ncbi.nlm.nih.gov/pubmed/11918817>
- Haijema BJ, Hahn J, Haynes J & Dubnau D (2001) A ComGA-dependent checkpoint limits growth during the escape from competence. *Mol Microbiol* **40**: 52–64 Available at: <http://www.ncbi.nlm.nih.gov/pubmed/11298275>
- Hamoen LW, Smits WK, de Jong A, Holsappel S & Kuipers OP (2002) Improving the predictive value of the competence transcription factor (ComK) binding site in *Bacillus subtilis* using a genomic approach. *Nucleic Acids Res* **30**: 5517–5528 Available at: <http://www.ncbi.nlm.nih.gov/pubmed/12490720>
- Hussain S, Wivagg CN, Szwedziak P, Wong F, Schaefer K, Izoré T, Renner LD, Holmes MJ, Sun Y, Bisson-Filho AW, Walker S, Amir A, Löwe J & Garner EC (2018) MreB filaments align along greatest principal membrane curvature to orient cell wall synthesis. *Elife* **7**: e32471 Available at: <https://elifesciences.org/articles/32471>
- Nicolas P, Mäder U, Dervyn E, Rochat T, Leduc A, Pigeonneau N, Bidnenko E, Marchadier E, Hoebeke M, Aymerich S, Becher D, Bisicchia P, Botella E, Delumeau O, Doherty G, Denham EL, Fogg MJ, Fromion V, Goelzer A, Hansen A, et al (2012) Condition-Dependent Transcriptome Reveals High-Level Regulatory Architecture in *Bacillus subtilis*. *Science* (80-.). **335**: 1103–1106 Available at: <http://www.sciencemag.org/content/335/6072/1103.abstract>
- Ogura M, Yamaguchi H, Kobayashi K, Ogasawara N, Fujita Y & Tanaka T (2002) Whole-genome analysis of genes regulated by the *Bacillus subtilis* competence transcription factor ComK. *J Bacteriol* **184**: 2344–2351 Available at: <http://www.ncbi.nlm.nih.gov/pubmed/11948146>
- Seto H & Tomasz A (1975) Selective release of a deoxyribonucleic acid binding factor from the surface of competent pneumococci. *J. Bacteriol.* **124**: 969–976
- Yadav T, Carrasco B, Serrano E & Alonso JC (2014) Roles of *bacillus subtilis* Dpra and Ssba in RecA-mediated genetic recombination. *J. Biol. Chem.* **289**: 27640–27652

REVIEWERS' COMMENTS:

Reviewer #1 (Remarks to the Author):

I am satisfied that the authors have responded well to my concerns. I think that they have developed and supported a useful working hypothesis that will potentially clarify an important step in the transformation of *B. subtilis*. Clearly their work lays the basis for much more fruitful investigation in the future.

Reviewer #2 (Remarks to the Author):

[No further comments for author.]

Reviewer #3 (Remarks to the Author):

With additional mutant analysis the manuscript was improved, however I still have the following major concerns

- 1) Lack of direct evidence to support there are two types of wall teichoic acids vWTA and cWTA . One can not excluded that cWTA is actually a new secondary cell wall polymer.
- 2) no direct evidence suggest that TuaH a WTA glycosyltransferase . Without enzymology data or additional mutant analysis, one cannot exclude that TuaH is a glycosyltransferase modify other second wall polymers and upregulated during competence

I understand that WTA analysis is challenging, tedious and time consuming and only a few labs have the expertise in WTA analysis and WTA glycosylation. However, I think additional mutant analysis would help explain how different is cWTA to vWTA , if they share something in common or if vWTA is a completely different second wall polymer. In the revised version, the authors described that deletion of WTA late genes like *dltA* and *tagE* has no effect on DNA binding , competence and transformation efficiency. As each WTA molecule is composed of a well conserved a linkage unit (with a disacccride GlcANc-ManNAc and two units of glycerolphosphate) and a main chain with alanyl and glucosyl modification , it will be beneficial to check how the mutations lead to the alteration of linkage unit and main chain may affect the DNA binding and transformation efficiency

I would suggest to check the following mutants in DNA binding and transformation efficiency

- 1) *tagE/dltA* double mutant and this mutant transformed with *tuaH* overexpressing vecor. This will tell if the main chain of the polyglycerolphosphate WTA is the substrate of *TuaH* and what are the function of the naked main chain in DNA binding and bacterial competence.

- 2) *tagA* mutant and *tuaH* over-expressing *tagA* mutant :
tagO and *tagA* are the only two early WTA genes that are dispensable and this experiments will tell if *tagA* mutant have the same phenotypes of *tagO* mutant in DNA binding and competence and if the cWTA has the same linkage unit as the vWTA.

Minor :

conA staining of the *dltA/tagO* double mutant:

in their response , the authors mentioned that the *conA* staining pattern of the mutant was not changed , same as the wild type .I am confused by the positive *conA* staining of this double mutant As the *conA* staining seems very specific to the Glc residues on WTA and this mutant does not have any WTA and no LTA alanylation , so I would expect this mutant would demonstrated a negative *ConA* staining .

Reviewer #1 (Remarks to the Author):

I am satisfied that the authors have responded well to my concerns. I think that they have developed and supported a useful working hypothesis that will potentially clarify an important step in the transformation of *B. subtilis*. Clearly their work lays the basis for much more fruitful investigation in the future.

We thank reviewer #1 for this positive final comment.

Reviewer #2 (Remarks to the Author):

[No further comments for author.]

We thank reviewer #2 for accepting the manuscript as it is.

Reviewer #3 (Remarks to the Author):

With additional mutant analysis the manuscript was improved, however I still have the following major concerns:

1) Lack of direct evidence to support there are two types of wall teichoic acids vWTA and cWTA . One cannot excluded that cWTA is actually a new secondary cell wall polymer.

We agree with reviewer #3 that we cannot exclude that cWTA represent a new secondary cell wall polymer that uses the same synthetic pathway than WTAs and thus is also inhibited by tunicamycin. This hypothesis, if confirmed in the future, would be exciting to explore.

2) no direct evidence suggest that TuaH a WTA glycosyltransferase. Without enzymology data or additional mutant analysis, one cannot exclude that TuaH is a glycosyltransferase modify other second wall polymers and upregulated during competence I understand that WTA analysis is challenging, tedious and time consuming and only a few labs have the expertise in WTA analysis and WTA glycosylation. However, I think additional mutant analysis would help explain how different is cWTA to vWTA , if they share something in common or if vWTA is a completely different second wall polymer.

In the revised version, the authors described that deletion of WTA late genes like *dltA* and *tagE* has no effect on DNA binding , competence and transformation efficiency. As each WTA molecule is composed of a well conserved a linkage unit (with a disacride GlcANc-ManNAc and two units of glycerolphosphate) and a main chain with alanyl and glucosyl modification , it will be beneficial to check how the mutations lead to the alteration of linkage unit and main chain may affect the DNA binding and transformation efficiency. I would suggest to check the following mutants in DNA binding and transformation efficiency:

1) *tagE/dltA* double mutant and this mutant transformed with *tuaH* overexpressing vector. This will tell if the main chain of the polyglycerolphosphate WTA is the substrate of *TuaH* and what are the function of the naked main chain in DNA binding and bacterial competence.

2) *tagA* mutant and *tuaH* over-expressing *tagA* mutant:

tagO and tagA are the only two early WTA genes that are dispensable and this experiments will tell if tagA mutant have the same phenotypes of tagO mutant in DNA binding and competence and if the cWTA has the same linkage unit as the vWTA.

We agree with reviewer #3 that overexpressing *tuaH* in diverse genetic backgrounds would help to confirm that the substrate of TuaH is polyglycerolphosphate WTA. However, these results would not be conclusive because of the pleiotropic effects associated to *tagE*, *dltA* and *tagA* deletions.

The experiment needed to conclusively confirm our hypotheses would be to link TuaH enzymatic activity to cWTA composition and to their ability to bind exogenous DNA. In addition to be challenging, tedious and time consuming experiments as indicated by reviewer #3, these experiments are beyond the scope of this manuscript. They will be developed in a following-up project

Minor :

conA staining of the *dltA/tagO* double mutant:

in their response , the authors mentioned that the conA staining pattern of the mutant was not changed , same as the wild type .I am confused by the positive conA staining of this double mutant As the conA staining seems very specific to the Glc residues on WTA and this mutant does not have any WTA and no LTA alanylation , so I would expect this mutant would demonstrated a negative ConA staining .

Oups, sorry! We don't understand why we mentioned the *dltA/tagO* double mutant in the revised version of our manuscript. Testing this double mutant was not requested by reviewer #3 in his/her precedent comments and we did not test it. This was certainly an editing mistake in the text and we apologize for it.

Since the ConA negative staining of a *tagE* mutant conclusively answers to the original comment of reviewer #3 there is no need to perform additional ConA staining experiments in other genetic backgrounds.